EMBO
Molecular Medicine

# CSF progranulin increases in the course of Alzheimer's disease and is associated with sTREM2, neurodegeneration and cognitive decline

Marc Suárez-Calvet[1,2,*,†] (iD), Anja Capell[1], Miguel Ángel Araque Caballero[3], Estrella Morenas-Rodríguez[1,4], Katrin Fellerer[1], Nicolai Franzmeier[3], Gernot Kleinberger[1,5] (iD), Erden Eren[1,6,7], Yuetiva Deming[8] (iD), Laura Piccio[9,10], Celeste M Karch[8,10,11], Carlos Cruchaga[8,10,11], Katrina Paumier[9,10,11], Randall J Bateman[9,10,11], Anne M Fagan[9,10,11], John C Morris[9,10,11], Johannes Levin[2,12], Adrian Danek[12], Mathias Jucker[13,14] (iD), Colin L Masters[15], Martin N Rossor[16], John M Ringman[17], Leslie M Shaw[18,19], John Q Trojanowski[18,19], Michael Weiner[20], Michael Ewers[3], Christian Haass[1,2,5,**] (iD), for the Dominantly Inherited Alzheimer Network[‡] & for the Alzheimer's Disease Neuroimaging Initiative[§]

## Abstract

Progranulin (PGRN) is predominantly expressed by microglia in the brain, and genetic and experimental evidence suggests a critical role in Alzheimer's disease (AD). We asked whether PGRN expression is changed in a disease severity-specific manner in AD. We measured PGRN in cerebrospinal fluid (CSF) in two of the best-characterized AD patient cohorts, namely the Dominant Inherited Alzheimer's Disease Network (DIAN) and the Alzheimer's Disease Neuroimaging Initiative (ADNI). In carriers of AD causing dominant mutations, cross-sectionally assessed CSF PGRN increased over the course of the disease and significantly differed from non-carriers 10 years before the expected symptom onset. In late-onset AD, higher CSF PGRN was associated with more advanced disease stages and cognitive impairment. Higher CSF PGRN was associated with higher CSF soluble TREM2 (triggering receptor expressed on myeloid cells 2) only when there was underlying pathology, but not in controls. In conclusion, we demonstrate that, although CSF PGRN is not a diagnostic biomarker for AD, it may together with sTREM2 reflect microglial activation during the disease.

**Keywords** Alzheimer's disease; biomarker; microglia; progranulin; TREM2
**Subject Categories** Biomarkers & Diagnostic Imaging; Neuroscience

1 Chair of Metabolic Biochemistry, Biomedical Center (BMC), Faculty of Medicine, Ludwig-Maximilians-Universität München, Munich, Germany
2 German Center for Neurodegenerative Diseases (DZNE) Munich, Munich, Germany
3 Institute for Stroke and Dementia Research, Klinikum der Universität München, Ludwig-Maximilians-Universität München, Munich, Germany
4 Department of Neurology, Institut d'Investigacions Biomèdiques, Hospital de la Santa Creu i Sant Pau, Universitat Autònoma de Barcelona, Barcelona, Catalonia, Spain
5 Munich Cluster for Systems Neurology (SyNergy), Munich, Germany
6 Izmir International Biomedicine and Genome Institute Dokuz, Eylul University, Izmir, Turkey
7 Department of Neuroscience, Institute of Health Sciences, Dokuz Eylul University, Izmir, Turkey
8 Department of Psychiatry, Washington University School of Medicine, St. Louis, MO, USA
9 Department of Neurology, Washington University School of Medicine, St. Louis, MO, USA
10 Hope Center for Neurological Disorders, Washington University in St. Louis, St. Louis, MO, USA
11 Knight Alzheimer's Disease Research Center, Washington University in St. Louis, St. Louis, MO, USA
12 Department of Neurology, Ludwig-Maximilians-Universität München, Munich, Germany
13 German Center for Neurodegenerative Diseases (DZNE) Tübingen, Tübingen, Germany
14 Department of Cellular Neurology, Hertie Institute for Clinical Brain Research, University of Tübingen, Tübingen, Germany
15 The Florey Institute of Neuroscience and Mental Health, University of Melbourne, Parkville, Vic., Australia
16 Dementia Research Centre, UCL Institute of Neurology, London, UK
17 Department of Neurology, Keck School of Medicine, University of Southern California, Los Angeles, CA, USA
18 Department of Pathology and Laboratory Medicine, Perelman School of Medicine, University of Pennsylvania, Philadelphia, PA, USA
19 Center for Neurodegenerative Disease Research, Institute on Aging, Perelman School of Medicine, University of Pennsylvania, Philadelphia, PA, USA
20 University of California at San Francisco, San Francisco, CA, USA
*Corresponding author. Tel: +49 89 4400 46549; Fax: +49 89 4400 46546; E-mail: msuarez@barcelonabeta.org
**Corresponding author. Tel: +49 89 4400 46549; Fax: +49 89 4400 46546; E-mail: christian.haass@mail03.med.uni-muenchen.de
‡https://dian.wustl.edu/
§Data used in preparation of this article were obtained from the Alzheimer's Disease Neuroimaging Initiative (ADNI) database (adni.loni.usc.edu). As such, the investigators within the ADNI contributed to the design and implementation of ADNI and/or provided data but did not participate in analysis or writing of this report. A complete listing of ADNI investigators can be found at: http://adni.loni.usc.edu/wp-content/uploads/how_to_apply/ADNI_Acknowledgement_List.pdf
†Present address: Barcelonaβeta Brain Research Center (BBRC), Pasqual Maragall Foundation, Barcelona, Catalonia, Spain

## Introduction

Haploinsufficiency of the gene encoding progranulin (PGRN) leads to frontotemporal lobar degeneration (FTLD) with TAR DNA binding protein 43 (TDP-43) deposition (FTLD-TDP) (Baker et al, 2006; Cruts et al, 2006; Neumann et al, 2006). Reduced PGRN protein in plasma, serum or cerebrospinal fluid (CSF) has been shown to be a reliable diagnostic biomarker for early detection of progranulin (GRN) mutation carriers (Ghidoni et al, 2008; Finch et al, 2009; Sleegers et al, 2009). Certain GRN variants may also increase the risk for Alzheimer's disease (AD) (Brouwers et al, 2008; Rademakers et al, 2008; Viswanathan et al, 2009; Lee et al, 2011; Cruchaga et al, 2012; Sheng et al, 2014; Xu et al, 2017), yet associations could not be confirmed in some other studies (Fenoglio et al, 2009; Mateo et al, 2013). Studies in AD mouse models indicated that PGRN is strongly increased in microglia clustering around amyloid plaques (Pereson et al, 2009) and may also affect amyloid β-peptide (Aβ) and tau deposition. PGRN may have beneficial effects on Aβ deposition, since PGRN deficiency in animal models increases Aβ deposition (Minami et al, 2014) and elevating PGRN expression reduces the amyloid plaque burden (Minami et al, 2014; Van Kampen & Kay, 2017). However, it has also been observed that PGRN deficiency leads to a decrease in diffuse Aβ plaque load (Takahashi et al, 2017; Hosokawa et al, 2018), which may point to a detrimental effect of PGRN in Aβ deposition. In regard to tau pathology, PGRN deficiency accelerates tau deposition and phosphorylation in human tau-expressing mice (Hosokawa et al, 2015; Takahashi et al, 2017). Furthermore, an AD-associated GRN variant (rs5848), which causes a decrease in PGRN levels in plasma and CSF (Rademakers et al, 2008; Nicholson et al, 2014; Morenas-Rodríguez et al, 2015), is associated with increased CSF T-tau levels in participants of the ADNI study (Takahashi et al, 2017). Together, these results indicate a protective role of PGRN against the development of tau pathology and/or neurodegeneration.

PGRN is a 593 amino acid protein that contains seven and a half tandem repeats forming the granulin domains (Bateman et al, 1990; Bhandari et al, 1992). It is targeted through the secretory pathway and secreted into the extracellular space. Secreted PGRN can be taken up and targeted to endosomes/lysosomes (Hu et al, 2010; Zhou et al, 2015b). Proteolytic processing of full-length PGRN may generate individual granulin peptides (Kleinberger et al, 2013; Holler et al, 2017). Within the brain, PGRN is predominantly expressed by microglia (Zhang et al, 2014; Lui et al, 2016; Chang et al, 2017), but some expression is also observed in neurons. Moreover, PGRN is significantly increased upon microglial activation (Daniel et al, 2000; Naphade et al, 2010; Petkau et al, 2010; Philips et al, 2010; Kleinberger et al, 2013; Suh et al, 2014). PGRN may be involved in the modulation of neuroinflammation since PGRN-deficient mice exhibit increased microglial activation and astrogliosis, as well as augmented expression of proinflammatory cytokines (Yin et al, 2009, 2010; Ahmed et al, 2010; Martens et al, 2012; Wils et al, 2012; Filiano et al, 2013; Minami et al, 2014). Whereas deficiency in the triggering receptor expressed on myeloid cells 2 (TREM2) locks microglia in a homeostatic stage (Mazaheri et al, 2017), loss of PGRN leads to their hyperactivation. Thus, deficiency of PGRN and TREM2 results in opposite functional deficits (Götzl et al, submitted).

Although genetic data and functional analyses in animal models provide some evidence for an involvement of PGRN in

AD, no changes in CSF PGRN in AD compared to healthy controls have been reported so far (Nicholson et al, 2014; Körtvélyessy et al, 2015; Morenas-Rodríguez et al, 2015; Wilke et al, 2017). However, these studies focused primarily on AD dementia and did not investigate the entire continuum of AD. Given that CSF biomarkers, such as the shed ectodomain of the triggering receptor expressed on myeloid cells 2 (sTREM2), show dynamic changes throughout the course of AD (Suárez-Calvet et al, 2016a,b), we aimed to cross-sectionally assess CSF PGRN changes at different stages of the disease. To this end, we studied a sample of persons at-risk for autosomal dominant AD (ADAD) recruited within the Dominantly Inherited AD Network (DIAN; http://dian.wustl.edu) (Bateman et al, 2012) and participants with late-onset AD from the Alzheimer's Disease Neuroimaging Initiative (ADNI; http://adni.loni.usc.edu; Weiner et al, 2012). We investigated whether CSF PGRN (i) increases in relation to the clinical course of AD; (ii) is associated with cognitive impairment and neuroimaging markers of neurodegeneration (fludeoxyglucose positron emission tomography, FDG-PET and hippocampal volume) in subjects with AD; and (iii) is associated with the microglial-derived protein sTREM2 as well as biomarkers of amyloid and tau pathology.

## Results

### CSF PGRN increases throughout the course of autosomal dominant Alzheimer's disease

Cross-sectionally, we assessed CSF PGRN in 215 participants from the DIAN initiative, including 130 mutation carriers (MC) and 85 non-carriers (NC; Table 1). All analyses described were adjusted for gender, age and APOE ε4 status, unless stated otherwise.

The levels of CSF PGRN were significantly increased in MC compared to NC ($F_{1,210} = 17.6$, $P = 0.00004$, Fig 1A, Appendix Table S1). In contrast to CSF sTREM2 (Henjum et al, 2016; Heslegrave et al, 2016; Piccio et al, 2016; Suárez-Calvet et al, 2016a,b), CSF PGRN was not significantly associated with age, neither in the entire sample ($\beta = 0.118$, $P = 0.079$), nor when stratifying by mutation status (for NC: $\beta = 0.125$, $P = 0.283$; for MC: $\beta = 0.131$, $P = 0.142$, Fig 1B). Consistent with previous publications (Nicholson et al, 2014; Morenas-Rodríguez et al, 2015), CSF PGRN was higher in males than in females ($F_{1,210} = 6.35$, $P = 0.012$, Fig 1C) and were not affected by APOE ε4 status ($F_{1,210} = 0.041$, $P = 0.840$). CSF PGRN did not differ between the three ADAD-associated genes (PSEN1, PSEN2 and APP) among the MC participants ($F_{2,124} = 0.77$, $P = 0.464$, Fig 1D).

We determined how CSF PGRN changes as a function of the estimated years from expected symptom onset (EYO) in MC compared to NC. Mean estimated levels of CSF PGRN at consecutive 5-year interval EYO were computed by a linear regression model including mutation status, EYO and gender as predictor variables (see Statistical analysis section). As described by Bateman et al (2012), we tested first-, second- and third-order EYO terms (EYO, $EYO^2$ and $EYO^3$, respectively) as well as their interaction with mutation status. The first-order (i.e. linear) model best fitted the data (Appendix Table S2). The interaction of EYO with mutation status

**Table 1.** DIAN participants' characteristics.

| | Autosomal dominant Alzheimer's disease (DIAN study) | | |
|---|---|---|---|
| | **Non-carriers (NC) (n = 85)** | **Mutation carriers (MC) (n = 130)** | **P-value (group effect)** |
| Age, years | 40.0 (10.8) | 39.9 (10.7) | 0.955 |
| Females, % | 55.3 | 49.2 | 0.384 |
| *APOE* ε4 carriers, % | 35.3 | 27.7 | 0.237 |
| Participant EYO, y | −6.81 (11.7) | −6.68 (10.7) | 0.931 |
| Education level, y | 14.8 (2.40) | 13.7 (3.19) | 0.005* |
| MMSE, scores | 29.0 (1.26) | 25.6 (6.16) | < 0.0001* |
| CSF biomarkers, pg/ml[a] | | | |
| T-tau | 60.1 (28.1) | 130 (97.0) | < 0.0001* |
| P-tau$_{181P}$ | 30.5 (10.3) | 69.8 (42.0) | < 0.0001* |
| Aβ$_{1–42}$ | 425 (136) | 322 (160) | < 0.0001* |
| sTREM2 | 2,796 (1,305) | 3,520 (1,640) | 0.0007* |
| PGRN[b] | 927 (174) | 1,041 (199) | 0.00004* |
| Cognitive status, %[c] | | | |
| CDR = 0 | 96.5 | 43.1 | |
| CDR = 0.5 | 3.5 | 39.2 | |
| CDR = 1 | 0 | 11.5 | |
| CDR = 2–3 | 0 | 6.1 | |
| Family mutations, %[d] | | | |
| *PSEN1* | 58.8 | 77.7 | |
| *PSEN2* | 14.1 | 8.5 | |
| *APP* | 27.1 | 13.8 | |

Aβ$_{1–42}$, amyloid-β 42; AD, Alzheimer disease; *APOE*, apolipoprotein E; APP, amyloid precursor protein; CSF, cerebrospinal fluid; EYO, estimated years from expected symptom onset; MMSE, Mini-Mental State Examination; *PSEN1*, presenilin 1; *PSEN2*, presenilin 2; P-tau$_{181P}$, tau phosphorylated at threonine 181; T-tau, total tau; y, years.

Data are expressed as mean (M) and standard deviation (SD) or percentage (%), as appropriate. Pearson's chi-square tests were used for the group comparisons of categorical variables and two-sample independent *t*-tests to compare continuous variables.

*Significant differences. The *P*-values indicated in the last column refer to the group effects in these tests.

[a]CSF core biomarkers in DIAN were measured using the Luminex bead-based multiplexed xMAP technology (INNO-BIA AlzBio3, Innogenetics).

[b]CSF PGRN differences were assessed by a linear model adjusted for age, gender and *APOE* ε4 status (see main text).

[c]Cognitive status was defined by the clinical dementia rating (CDR) score (0, cognitively normal; 0.5, very mild; 1, mild; 2, moderate; 3, severe dementia).

[d]In NC participants, the mutation present in their family is shown.

was a significant predictor of CSF PGRN changes ($P = 0.041$), indicating that the changes in CSF PGRN as a function of EYO differ between MC and NC. We compared the estimated CSF PGRN at 5-year EYO intervals by *t*-tests and observed that CSF PGRN started to be significantly increased in MC compared to NC at EYO = −10 (Table 2, Fig 2A, Appendix Table S3).

Following the approach of previous DIAN studies (Bateman *et al*, 2012; Fagan *et al*, 2014; Suárez-Calvet *et al*, 2016a,b), we added CSF PGRN in the temporal sequence of biomarker changes within the course of ADAD (Fig 2B). Like CSF sTREM2, changes in CSF PGRN occurred after alterations in markers for brain amyloidosis and neuronal injury, as measured by CSF T-tau.

Next, we compared CSF PGRN in MC at different clinical stages defined by the clinical dementia rating (CDR) score with that of the NC. We hence compared four groups: NC, cognitively normal (CDR = 0) MC, very mild dementia (CDR = 0.5) MC and mild-to-severe dementia (CDR > 1) MC. Participants with a CDR ≥ 1 were grouped together because of the low number of subjects in these groups (Table 1). We conducted an analysis of covariance

(ANCOVA) adjusted for age, gender, *APOE* ε4 status and education, and we observed a significant difference between the four groups ($F_{3,207} = 5.77$, $P = 0.001$). A least significant difference (LSD) pairwise *post hoc* comparisons revealed that all MC groups (CDR = 0: M = 1002, SD = 186; CDR = 0.5: M = 1072, SD = 201; CDR ≥ 1: M = 1065, SD = 214; pg/ml) had significantly higher CSF PGRN than the NC (M = 927 pg/ml, SD = 174; $P = 0.026$, $P = 0.0001$ and $P = 0.018$, respectively). No differences were found between the MC groups at different clinical stages. Likewise, CSF PGRN levels were not associated with Mini-Mental State Examination (MMSE; $P = 0.881$) or CDR sum of boxes (CDR-SB; $P = 0.812$) among MC.

**CSF PGRN increases throughout the course of late-onset Alzheimer's disease**

To further validate our findings in subjects with late-onset AD, we studied a total of 1,017 participants of the ADNI study (demographics of the entire ADNI sample are summarized in Appendix Table S4). In the entire sample, and consistent with the

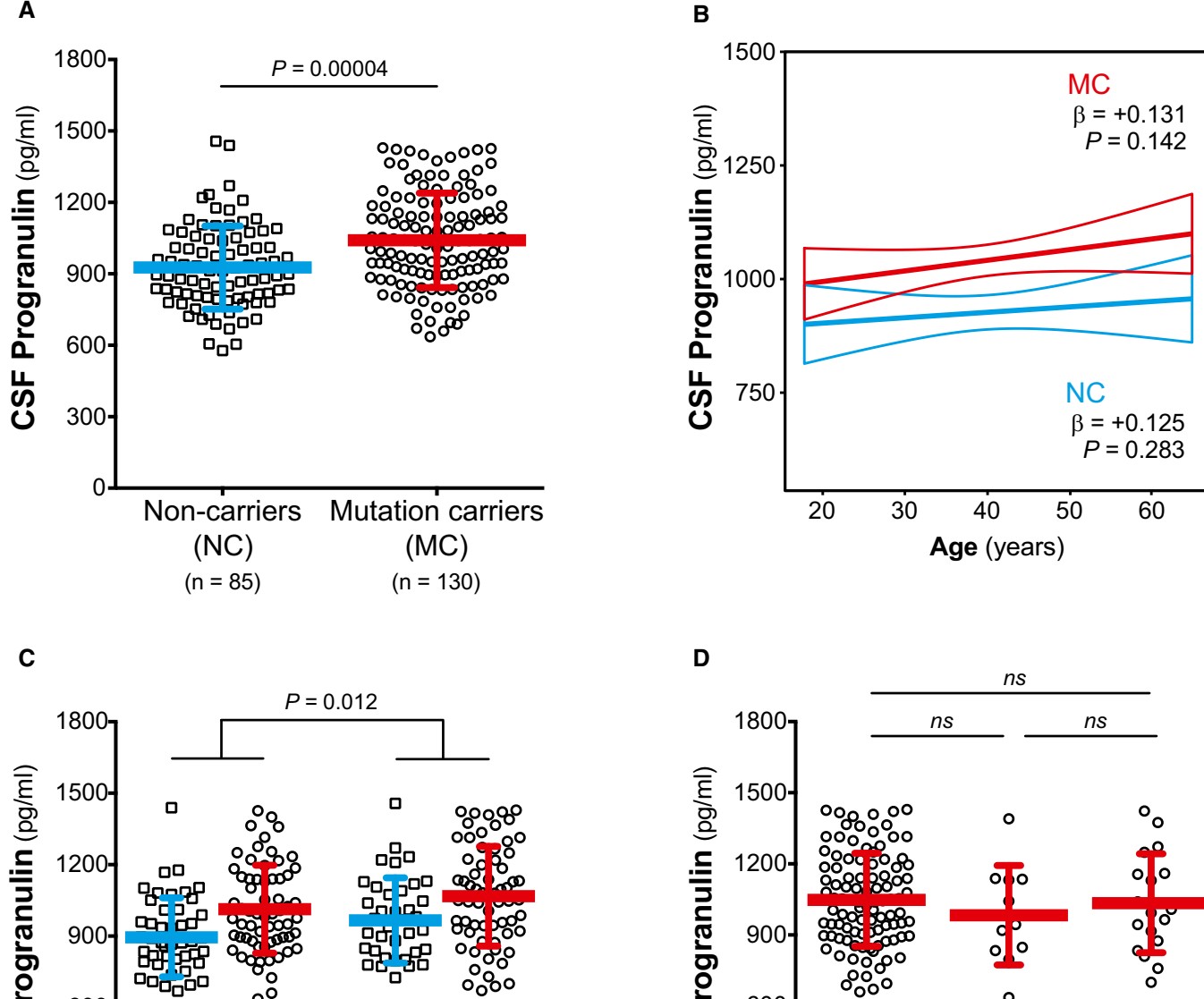

**Figure 1. Association of CSF PGRN with mutation status, age and gender.**

A  CSF PGRN is increased in mutation carriers (MC) compared to non-carriers (NC).
B  CSF PGRN is not associated with age in either NC or MC.
C  CSF PGRN is increased in males compared to females.
D  CSF PGRN levels do not differ among MC participants carrying a *PSEN1*, *PSEN2* or *APP* mutation.

Data information: The blue or red bars in (A), (C) and (D) represent the mean and the standard deviation (SD). Group comparisons were assessed by a linear model adjusting by age, gender and *APOE* ε4 status. The solid lines in (B) indicate the regression line for each of the groups and the 95% confidence interval (CI) calculated by a linear model adjusting by gender and *APOE* ε4 status. The standardized regression coefficients (β) and the *P*-values are also shown. In graph (B), the individual values are not shown in order to protect participants' confidentiality. All analysis and graphs are performed excluding 3 PGRN values outliers. Including the outliers in the analysis rendered similar results (Appendix Table S1). *APP*, amyloid precursor protein; CSF, cerebrospinal fluid; ns, non-significant; *PSEN1*, presenilin 1; *PSEN2*, presenilin 2.

                    

**Table 2.  CSF PGRN estimates (pg/ml) in MCs and NCs as a function of EYO.**

| | Estimated years from expected symptom onset (EYO) | | | | | | | |
|---|---|---|---|---|---|---|---|---|
| | −25 | −20 | −15 | −10 | −5 | 0 | +5 | +10 |
| Non-carriers | 974 | 978 | 983 | 988 | 992 | 997 | 1,002 | 1,006 |
| Mutation carriers | 992 | 1,022 | 1,052 | 1,082 | 1,112 | 1,142 | 1,172 | 1,202 |
| Difference | 18 | 44 | 69 | 94 | 120 | 145 | 170 | 196 |
| 95% CI | [−104, 140] | [−64, 151] | [−27, 165] | [4, 184] | [29, 210] | [48, 242] | [62, 279] | [72, 319] |
| *P*-value | 0.768 | 0.423 | 0.159 | 0.040* | 0.010* | 0.004* | 0.002* | 0.002* |

CI, confidence interval.
Mean estimated levels of CSF PGRN were obtained by a linear model including gender, mutation status, EYO and the interaction between mutation status and EYO as covariates (see Statistical analysis section and Appendix Table S2). For each EYO, the group difference, 95% CI and the *P*-value for the two-sample independent *t*-test are reported. Participants with EYO > +20 (1 NC and 2 MC) were excluded from the analysis. The same analysis including the PGRN outliers and those participants with an EYO > +20 yielded identical results (Appendix Table S3). Differences are calculated from unrounded values.
*Significant difference.

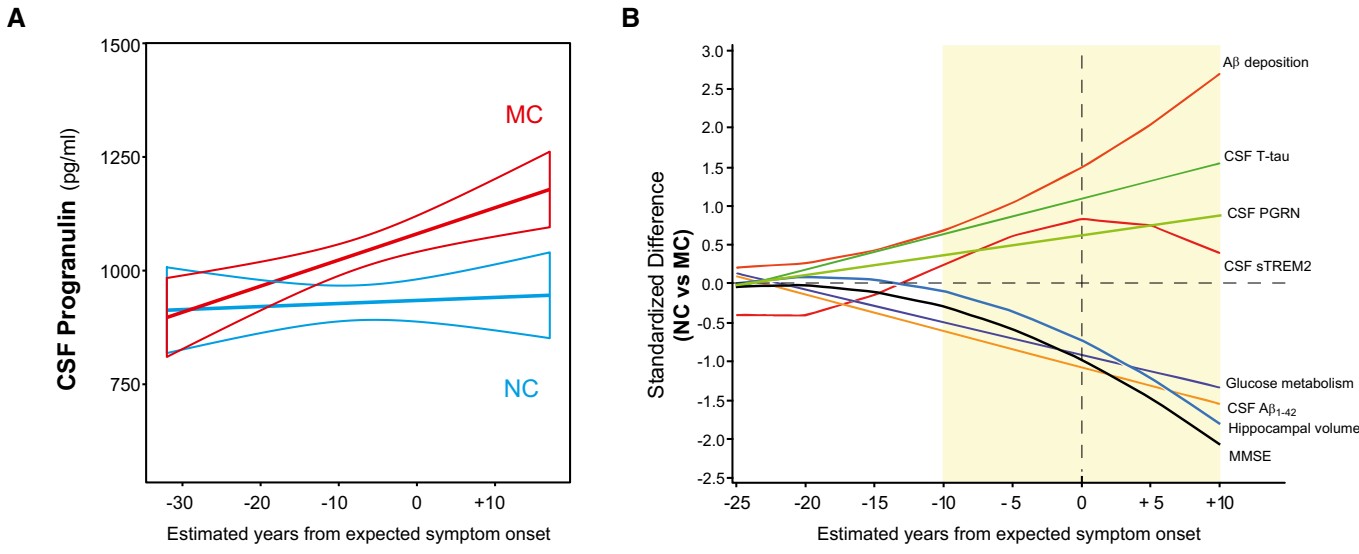

**Figure 2.  Changes in CSF PGRN as a function of EYO.**

A   CSF PGRN as a function of EYO in mutation carriers (MC, red) and non-carriers (NC, blue). The solid lines indicate the regression line for each of the groups and the 95% confidence interval (CI) calculated by a linear model adjusting by gender. The interaction term of mutation status and EYO is significant (*P* = 0.041), also when including PGRN outliers and participants with EYO > +20 (*P* = 0.030). Individual data points are not displayed to prevent disclosure of mutation status.

B   The graph depicts the standardized differences in CSF PGRN between MCs and NCs as a function of EYO, in the context of other biomarker and cognitive changes. The curves were generated by the linear model that best fit each marker (see Statistical analysis section and Appendix Table S2). CSF PGRN is significantly increased in MC compared to NC 10 years before the expected symptom onset (shadowed area) after brain amyloidosis and brain injury (as measured by CSF T-tau) have started, and shortly before CSF sTREM2 starts to increase.

Data information: Aβ$_{1-42}$: amyloid-β 42; CSF, cerebrospinal fluid; MC, mutation carrier; MMSE, Mini-Mental State Examination; NC, non-carrier; T-tau, total tau.

findings in ADAD, CSF PGRN was higher in males than females ($F_{1,1013}$ = 26.4, $P < 0.0001$, Appendix Table S5). A trend towards lower PGRN was observed in *APOE* ε4 carriers ($F_{1,1013}$ = 3.40, $P = 0.066$), but was not associated with age (β = +0.049, $P = 0.118$).

In order to study the changes in CSF PGRN at different stages of disease severity, we defined disease stage according to the recently proposed biomarker-based A/T/N framework (Jack *et al*, 2016a) in combination with the CDR score (Morris, 1993; Table 3). The A/T/N classification is the basis of the 2018 National Institute on Aging–Alzheimer's Association (NIA-AA) Research Framework (Jack *et al*, 2018) and defines three binary biomarker categories: (i) aggregated Aβ (A+/A−), (ii) aggregated tau (T+/T−) and (iii)

neurodegeneration or neuronal injury (N+/N−). Herein, each of these categories was defined using the AD CSF core biomarkers, namely CSF Aβ$_{1-42}$ (A), P-tau$_{181P}$ (T) and T-tau (N). The aggregated tau (T) and neurodegeneration (N) groups were merged together to simplify the number of groups to compare (only 5.3% of the participants of the ADNI total sample displayed discrepancies between the T and N biomarker groups; see Materials and Methods section for a comprehensive description on the classification). We also classified the participants based on their clinical status (C), as measured by the well-established CDR global score (Morris, 1993), into cognitively unimpaired (CDR = 0), very mild dementia (CDR = 0.5) and mild dementia (CDR = 1). The combination of both the biomarker

**Table 3.  Classification of ADNI participants based on their biomarker profile and clinical stage.**

| | | Clinical stage (C) | | |
| --- | --- | --- | --- | --- |
| | | CDR = 0 (cognitively unimpaired) | CDR = 0.5 (very mild dementia) | CDR = 1 (mild dementia) |
| Biomarker profile | A−/TN− | Healthy controls n = 128 | n = 120 | n = 2 |
| | A+/TN− | Preclinical AD A+/TN− n = 56 | n = 96 | n = 15 |
| | A+/TN+ | Preclinical AD A+/TN+ n = 48 | AD CDR = 0.5 n = 289 | AD CDR = 1 n = 81 |
| | A−/TN+ | n = 74 | n = 100 | n = 8 |
| | | suspected non-Alzheimer's pathophysiology (SNAP) | | |

A, amyloid-β biomarker status; AD, Alzheimer's disease; CDR, clinical dementia rating; N, neurodegeneration biomarker status; T, tau pathology biomarker status.

ADNI participants were classified based on their CSF biomarker profile and their clinical stage, which yielded 12 different categories. Columns depict the clinical stage (C) as defined by the clinical dementia rating (CDR) scale. Rows depict the biomarker profiles. Each of the three biomarker groups (A/T/N) was binarized into positive or negative (+/−). T and N were merged to simplify the classification: TN− indicates that both T and N biomarkers are normal, and TN+ indicates that T and/or N biomarkers are abnormal.

The grey highlighting indicates the grouping used for comparisons in the main text. Light grey highlights the healthy controls (n = 128), middle grey the groups included in the Alzheimer's *continuum* (n = 474) and dark grey the suspected non-Alzheimer's pathophysiology (SNAP) group (n = 182). Bold text indicates the groups analysed in the main analysis, namely "healthy controls", "Preclinical AD A+/TN−", "Preclinical AD A+/TN+", "AD CDR = 0.5" and "AD CDR = 1".

and the clinical classification rendered twelve different groups that are summarized in Table 3. After classifying the participants of the ADNI sample, we analysed the data following two approaches. In a first approach, we attempted to model disease stages within the Alzheimer's *continuum* as a combination of biomarkers and clinical symptoms similar to what was proposed by the previous 2011 NIA-AA diagnostic criteria (Albert *et al*, 2011; McKhann *et al*, 2011; Sperling *et al*, 2011), and it was done in order to render the disease staging in late-onset AD more comparable to that in ADAD defined by EYO. Second, in an exploratory approach, we compared PGRN levels between the different A/T/N categories within each clinical stage.

Thus, we first asked whether CSF PGRN increases in relation to the biomarker-defined clinical stages, and hence parallels PGRN changes as a function of EYO in ADAD. For this purpose, we compared the "healthy control" group (highlighted in light grey in Table 3) with those groups that belong to the Alzheimer's *continuum* (highlighted in middle grey in Table 3). We hence compared five groups in this first analysis, namely (i) "healthy controls", (ii) "Preclinical AD A+/TN−" (iii) "Preclinical AD A+/TN+" (iv) "AD CDR = 0.5" and (v) "AD CDR = 1".

The demographics and clinical features of these groups are summarized in Table 4. We conducted an ANCOVA controlling for age, gender and *APOE* ε4 status, and we found that CSF PGRN significantly differed between groups ($F_{4,594} = 7.32$, $P < 0.0001$). Bonferroni corrected pair-wise *post hoc* comparisons indicated that CSF PGRN was significantly higher in the "AD CDR = 1" group compared to the "healthy controls" ($P = 0.006$) and "Preclinical AD A+/TN−" ($P < 0.0001$) groups (Fig 3 and Table 4). Interestingly, the "Preclinical AD A+/TN−" group had significantly lower CSF

PGRN than the rest of the Alzheimer's *continuum* groups (Fig 3 and Table 4) but not to "healthy controls". No other group differences were found.

Given that *GRN rs5848* is a well-known modifier of PGRN levels (Rademakers *et al*, 2008; Nicholson *et al*, 2014; Morenas-Rodríguez *et al*, 2015), we repeated the same analysis but also accounting for the *rs5848* genotype, which was available for 58.5% of the cases (see Table 4). CSF PGRN still significantly differed between groups ($F_{4,342} = 5.66$, $P = 0.0002$), where significant differences were observed when "AD CDR = 1" was compared to "healthy controls" ($P = 0.030$) and "Preclinical AD A+/TN−" ($P = 0.0001$) groups in Bonferroni corrected pair-wise *post hoc* comparisons. Similarly, CSF PGRN remained lower in "Preclinical AD A+/TN−" compared to "AD CDR = 0.5" ($P = 0.002$) and "AD CDR = 1" ($P = 0.0001$) groups and a tendency existed when compared to "Preclinical AD A+/TN+" ($P = 0.082$). Consistent with previous reports (Rademakers *et al*, 2008; Nicholson *et al*, 2014; Morenas-Rodríguez *et al*, 2015), there was a significant effect of the *GRN rs5848* genotype on CSF PGRN levels ($F_{2,342} = 20.6$, $P < 0.0001$), such that the CSF PGRN mean level of the *GRN rs5848* TT carriers (M = 1348 pg/ml, SD = 330) was significantly lower than that of the *rs5848* CT carriers (M = 1524 pg/ml, SD = 277, $P = 0.0001$) and *rs5848* CC carriers (M = 1659 pg/ml, SD = 379, $P < 0.0001$) groups (Bonferroni corrected pair-wise *post hoc* comparisons). Together, the CSF PGRN changes across biomarker-defined clinical stages are consistent with the increase we found in ADAD throughout EYO.

To further confirm that CSF PGRN changes across the disease course, we also tested whether CSF PGRN levels are associated with cognitive and functional scores in those participants that fall in the Alzheimer's *continuum* category (Table 5; Fig 4). Our primary cognitive measures were the composite scores ADNI-Mem (Fig 4A), for memory performance, and ADNI-EF (Fig 4B), for executive function, since they have been previously validated in the ADNI study, are robust and have external validity (Crane *et al*, 2012; Gibbons *et al*, 2012; Habeck *et al*, 2012). We computed three linear regression models, including as main predictors: unadjusted (Model 1); adjusted for age, gender, *APOE* ε4 status and education (Model 2); and additionally adjusted for CSF Aβ$_{1-42}$ and CSF T-tau (Model 3, Table 5). Higher CSF PGRN was associated with lower memory performance as measured by ADNI-Mem (β = −0.145, $P = 0.002$) and executive function as assessed by ADNI-EF (β = −0.145, $P = 0.002$) in an unadjusted model (Model 1, Table 5 and Appendix Table S6). These associations remained significant after adjustment for age, gender, *APOE* ε4 status and education (Model 2, ADNI-Mem: β = −0.140, $P = 0.002$, Fig 4A; ADNI-EF: β = −0.150, $P = 0.0008$, Fig 4B; Table 5 and Appendix Table S6). We also studied ADAS-Cog 11, ADAS-Cog 13, MMSE and CDR-SB as secondary cognitive measures, and the results were similar (Fig 4C–F, Table 5 and Appendix Table S6). The associations remain significant after accounting for Aβ$_{1-42}$ and T-tau (Model 3) for ADNI-EF, ADAS-Cog 11 and ADAS-Cog 13, and a tendency existed for ADNI-Mem and MMSE (Table 5, Appendix Table S6), indicating that the association between CSF PGRN and cognitive scores is, at least in part, not dependent on the AD CSF core biomarkers.

We also tested whether CSF PGRN is associated with neuroimaging biomarkers of neurodegeneration typically affected in AD, namely temporo-parietal FDG-PET uptake and total hippocampal volume, in the Alzheimer's *continuum* category (Fig 5). In a linear

**Table 4. ADNI participants' characteristics for the control and Alzheimer's *continuum* groups in ADNI.**

| | Healthy controls (n = 128) | Alzheimer's *continuum* (n = 474) | | | | P-value (group effect) |
|---|---|---|---|---|---|---|
| | | Preclinical AD A+/TN− (n = 56) | Preclinical AD A+/TN+ (n = 48) | AD CDR = 0.5 (n = 289) | AD CDR = 1 (n = 81) | |
| Age, years | 72.5 (5.37) | 73.2 (5.96) | 76.5 (5.37) | 73.3 (7.05) | 73.8 (9.27) | 0.017* |
| Female, % | 48.4 | 46.4 | 52.1 | 42.6 | 50.6 | 0.542 |
| *APOE* ε4 carriers, % | 14.8 | 39.3 | 58.3 | 76.5 | 74.1 | < 0.0001* |
| *GRN rs5848* TT carriers, %[a] | 5.20 | 13.2 | 6.90 | 13.6 | 14.8 | 0.472 |
| Education, y | 16.3 (2.79) | 16.5 (2.74) | 16.5 (2.47) | 15.9 (2.87) | 15.1 (2.78) | 0.010* |
| Cognitive tests, scores | | | | | | |
| ADNI-Mem | 1.14 (0.59) | 1.04 (0.61) | 0.86 (0.56) | −0.28 (0.64) | −1.02 (0.46) | < 0.0001* |
| ADNI-EF | 0.94 (0.75) | 0.74 (0.70) | 0.40 (0.61) | −0.12 (0.79) | −1.07 (0.78) | < 0.0001* |
| ADAS-Cog11 | 5.96 (3.12) | 5.85 (3.06) | 6.54 (3.01) | 13.2 (5.34) | 23.2 (6.98) | < 0.0001* |
| ADAS-Cog13 | 8.94 (4.57) | 9.27 (4.51) | 10.2 (4.45) | 21.2 (7.38) | 34.6 (7.95) | < 0.0001* |
| MMSE | 29.1 (1.12) | 28.9 (1.23) | 29.0 (1.20) | 26.3 (2.39) | 22.7 (2.06) | < 0.0001* |
| CDR-SB | 0.016 (0.09) | 0.054 (0.16) | 0.073 (0.21) | 2.09 (1.09) | 5.62 (1.17) | < 0.0001* |
| CSF biomarkers, pg/ml[b] | | | | | | |
| T-tau | 184 (31.8) | 166 (41) | 329 (77.5) | 382 (135) | 395 (137) | < 0.0001* |
| P-tau$_{181P}$ | 16.3 (2.87) | 15.4 (4.09) | 33.0 (8.93) | 39.1 (14.5) | 39.4 (15.1) | < 0.0001* |
| Aβ$_{1-42}$ | 1,455 (227) | 724 (193) | 712 (173) | 636 (167) | 575 (159) | < 0.0001* |
| PGRN[c] | 1,502 (279) | 1,394 (365) | 1,569 (305) | 1,541 (343) | 1,649 (375) | < 0.0001* |

A, amyloid-β biomarker status; Aβ$_{1-42}$, amyloid-β 42; AD, Alzheimer disease; APOE, apolipoprotein E; ADAS-Cog, Alzheimer's disease Assessment Scale—cognitive subscale; ADNI-Mem, ADNI memory composite score; ADNI-EF, ADNI executive function composite score; CDR, clinical dementia rating; CDR-SB, clinical dementia rating sum of boxes; CSF, cerebrospinal fluid; MMSE, Mini-Mental State Examination; N, neurodegeneration biomarker status; P-tau$_{181P}$, tau phosphorylated at threonine 181; T, tau pathology biomarker status; T-tau, total tau; y, years.
Data are expressed as mean (M) and standard deviation (SD) or percentage (%), as appropriate. Pearson's chi-square tests were used for the group comparisons of categorical variables and one-way ANOVA to compare continuous variables.
*Significant differences. The *P*-values indicated in the last column refer to the group effects in these tests.
[a]*GRN rs5848* genotype was available in 77 "healthy controls" (60%), 38 "Preclinical AD A+/TN−" (68%), 29 "Preclinical AD A+/TN+" (60%), 154 "AD CDR = 0.5" (53%) and 54 "AD CDR = 1" (67%).
[b]The CSF core biomarker measurements were performed using the electrochemiluminescence immunoassays, total-tau CSF, phospho-tau(181P) CSF and Elecsys β-amyloid(1–42) CSF. The Elecsys β-amyloid(1–42) assay has an upper technical limit of 1700 pg/ml; the values above this limit were truncated to this value.
[c]CSF PGRN levels were log-transformed and assessed by a linear model adjusted for age, gender and *APOE* ε4 status (see main text).

regression analysis adjusted for age, gender, *APOE* ε4 status and education, higher CSF PGRN was associated with lower temporo-parietal FDG-PET uptake (β = −0.133, *P* = 0.010) (Fig 5A), but not total hippocampal volume (β = −0.063, *P* = 0.151, Fig 5B).

In the second approach, we examined CSF PGRN between biomarker profile [as defined by the A/T/N classification (Jack *et al*, 2016a)] within each clinical stage (Fig EV1A). Unlike the former analysis of the Alzheimer's *continuum*, this is an unbiased comparison that does not assume any particular sequence of biomarkers in the course of the disease. Appendix Table S4 displays a summary of the demographics of the entire ADNI sample, analysed here. As expected, the CDR = 1 clinical stage had some biomarker profiles with low number of subjects that preclude any comparisons but are shown in Fig EV1A for the sake of completeness. Following this approach, we found that the "A−TN+" biomarker profiles (i.e. SNAP) had the highest CSF PGRN, which was significantly higher than the rest of the biomarker profiles within the same clinical stage (Fig EV1A). On the contrary, the "A+TN−" biomarker profiles had the lowest CSF PGRN. We next grouped the subjects that fall into each of the biomarker category of "suspected non-Alzheimer's

pathophysiology" (SNAP; Table 3, highlighted in dark grey) (Jack *et al*, 2012, 2016b; Caroli *et al*, 2015; Dani *et al*, 2017) and we compared them to the healthy controls and the Alzheimer's *continuum* category as a whole. CSF PGRN significantly differ between categories ($F_{2,778}$ = 24.7, *P* < 0.0001, Fig EV1B), and Bonferroni corrected pair-wise *post hoc* tests showed that CSF PGRN was significantly higher in the SNAP category compared to the healthy controls (*P* < 0.0001) and Alzheimer's *continuum* categories (*P* < 0.0001). Finally, we also tested the association between CSF PGRN and cognitive function and neuroimaging biomarkers within the SNAP category, with the same linear regression models we applied in the analysis focused on the Alzheimer's *continuum* category. Strikingly, and in contrast with the results in the Alzheimer's *continuum*, CSF PGRN was not associated with cognitive decline in the SNAP group in any of the models tested (Appendix Table S7). In line with this finding, CSF PGRN was also not associated with temporo-parietal FDG-PET uptake in the SNAP group (β = +0.102, *P* = 0.203). These results suggest that, despite the fact that CSF PGRN is increased in SNAP, it selectively associates with disease severity in AD.

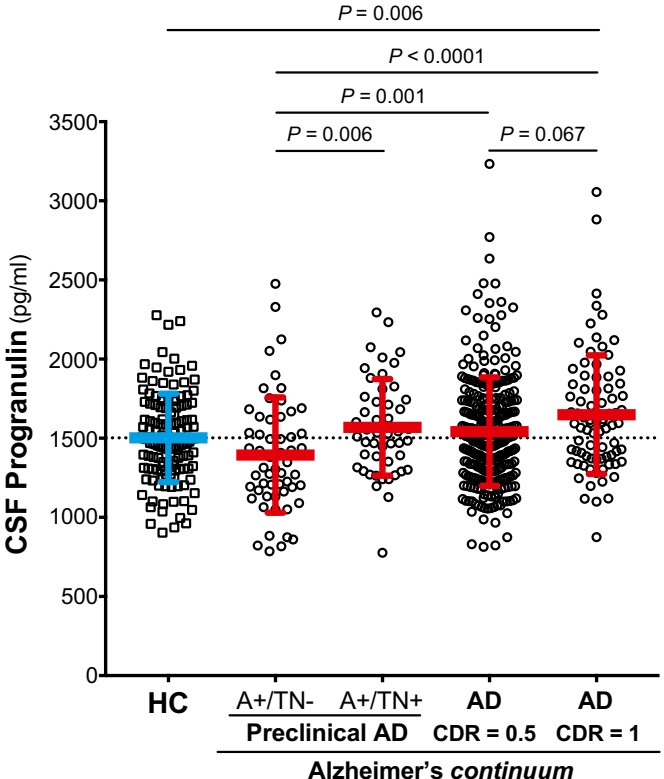

**Figure 3.  CSF PGRN levels across the Alzheimer's *continuum*.**

Scatter plot representing the levels of CSF PGRN in healthy controls (depicted in blue) and the different stages of the Alzheimer's *continuum* (depicted in red). The blue and the red bars represent the mean and the standard deviation (SD). The analysis and graphs were performed excluding CSF PGRN outliers (1 "healthy control", 1 "Preclinical AD A+/TN−", 4 "AD CDR = 0.5" and 1 "AD CDR = 1"). Including them yielded a similar result, and CSF PGRN was still significantly higher in the "AD CDR = 1" group compared to the "healthy controls" ($P = 0.001$) and "Preclinical AD A+/TN−" ($P = 0.0001$) groups. $P$-values were assessed by a one-way analysis of covariance adjusted for age, gender and *APOE* ε4, followed by Bonferroni corrected pair-wise *post hoc* comparisons. A: amyloid-β biomarker status; AD: Alzheimer's disease; CDR: clinical dementia rating; CSF, cerebrospinal fluid; N, neurodegeneration biomarker status; T: tau pathology biomarker status.

## CSF PGRN is not a clinical diagnostic biomarker for AD

We assessed the diagnostic accuracy of CSF PGRN to discriminate between AD and controls. To this regard, a receiver operating characteristic (ROC) curve analysis was undertaken (Fig EV2). The area under the curve (AUC) was 0.655 (95% CI 0.581–0.729, $P = 0.0001$, Fig EV2A) for discriminating ADAD mutation carriers from non-carriers of the DIAN study, and 0.607 (95% CI 0.528–0.686, $P = 0.009$, Fig EV2B) for discriminating late-onset AD CDR = 1 from healthy controls. Although the significant $P$-values denote that the AUC of CSF PGRN is significantly different from the area under the diagonal, which corresponds to a random performance of a test, these are low AUC that indicate a poor accuracy to discriminate between AD and controls. Together with the fact that CSF PGRN considerably overlaps between groups, these results show that, consistent with previous data (Nicholson *et al*, 2014; Körtvélyessy *et al*, 2015; Morenas-Rodríguez *et al*, 2015; Wilke *et al*, 2017), CSF PGRN is not useful as a diagnostic marker in AD.

## CSF PGRN is associated with CSF sTREM2 and markers of neurodegeneration in both autosomal dominant and late-onset Alzheimer's disease

We and others previously described an increase in the microglial-derived protein sTREM2 in the CSF of both ADAD and late-onset AD patients (Heslegrave *et al*, 2016; Piccio *et al*, 2016; Suárez-Calvet *et al*, 2016a,b). Since CSF PGRN also gradually increases during the course of the disease and it is mainly produced by activated microglia (Daniel *et al*, 2000; Naphade *et al*, 2010; Petkau *et al*, 2010; Philips *et al*, 2010; Kleinberger *et al*, 2013; Suh *et al*, 2014), we investigated whether these two proteins are associated (Figs 6A and B, and 7A–C). In a linear regression model, we found that CSF PGRN and CSF sTREM2 were significantly associated in ADAD mutation carriers of the DIAN study (β = +0.514, $P < 0.0001$, Fig 6B). Similarly, CSF PGRN and sTREM2 were also significantly associated in individuals of the Alzheimer's *continuum* (β = +0.344, $P < 0.0001$, Fig 7B) and the SNAP category (β = +0.296, $P < 0.0001$, Fig 7C) of the ADNI study. Importantly, these associations only occurred in AD and SNAP since no association was found in the NC of the DIAN study (β = +0.094, $P = 0.424$, Fig 6A) and in the healthy controls of the ADNI study (β = +0.106, $P = 0.246$, Fig 7A). Together, these results suggest that, whenever there is neurodegeneration (triggered by amyloidosis or other causes), there is a concurrent release of the microglial proteins PGRN and sTREM2 into the CSF.

We also tested the associations between CSF PGRN and each of the core CSF biomarkers of AD (T-tau, P-tau$_{181P}$ and Aβ$_{1–42}$) in linear regression models for both ADAD (DIAN) and late-onset AD (ADNI) (Figs 6C–H and 7D–L; Appendix Table S8 summarizes the results including biomarker outliers). The results paralleled those we previously found for CSF sTREM2 (Suárez-Calvet *et al*, 2016a, b). CSF PGRN was associated with CSF T-tau and CSF P-tau$_{181P}$, markers of neurodegeneration and neurofibrillary tangle degeneration, respectively, in both ADAD mutations carriers (T-tau: β = +0.258 $P = 0.008$; P-tau$_{181P}$: β = +0.191, $P = 0.041$; Fig 6D and F) and NC (T-tau: β = +0.240 $P = 0.032$; P-tau$_{181P}$: β = +0.290, $P = 0.008$, Fig 6C and E). Interestingly, this association was only present in the ADNI subjects of the Alzheimer's *continuum* (T-tau: β = +0.295 $P < 0.0001$; P-tau$_{181P}$: β = +0.280, $P < 0.0001$; Fig 7E and H), but not in healthy controls (T-tau: β = +0.157 $P = 0.087$; P-tau$_{181P}$: β = +0.118, $P = 0.195$; Fig 7D and G) or SNAP subjects (T-tau: β = +0.043 $P = 0.564$; P-tau$_{181P}$: β = +0.104, $P = 0.172$; Fig 7F and I). These findings are consistent with the fact that CSF PGRN is only associated with cognitive impairment in the Alzheimer's *continuum* category but not in SNAP and support the idea that CSF PGRN parallels the burden of the disease specifically in AD.

In contrast, we did not observe an association between CSF PGRN and Aβ$_{1–42}$ in either NC or MC of the DIAN sample (Fig 6G and H) and only a weak association was found in the Alzheimer's *continuum* and SNAP participants of the ADNI sample (Fig 7J–L).

## Discussion

In this cross-sectional study, we found that CSF PGRN increases throughout the course of AD, both in ADAD and in late-onset AD. Furthermore, CSF PGRN is associated with the microglial-derived

**Table 5.  Associations of CSF PGRN with cognitive measures in the Alzheimer's *continuum* category (ADNI sample).**

|  | Model 1 (unadjusted) | | Model 2 (adjusted for age, gender, *APOE* ε4 and education) | | Model 3 (also adjusted for Aβ$_{1-42}$ and T-tau) | |
|---|---|---|---|---|---|---|
|  | β | *P* | β | *P* | β | *P* |
| ADNI-Mem | −0.145 | 0.002* | −0.140 | 0.002* | −0.080 | 0.064 |
| ADNI-EF | −0.145 | 0.002* | −0.150 | 0.0008* | −0.116 | 0.001* |
| ADAS-Cog 11 | +0.138 | 0.003* | +0.135 | 0.003* | +0.095 | 0.036* |
| ADAS-Cog 13 | +0.149 | 0.001* | +0.148 | 0.001* | +0.108 | 0.016* |
| MMSE | −0.126 | 0.006* | −0.126 | 0.006* | −0.087 | 0.052 |
| CDR-SB | +0.174 | 0.011* | +0.125 | 0.007* | +0.070 | 0.119 |

Aβ$_{1-42}$, amyloid-β 42; ADAS-Cog, Alzheimer's disease Assessment Scale—cognitive subscale; ADNI-Mem, ADNI memory composite score; ADNI-EF, ADNI executive function composite score; CDR-SB, clinical dementia rating sum of boxes; MMSE, Mini-Mental State Examination; T-tau, total tau.

Associations between CSF PGRN and cognitive measures were studied only in the participants of the Alzheimer's *continuum* (n = 474) and were assessed by three different linear regression models. The standardized regression coefficients (β) and the *P*-values are shown.

Note that higher levels of CSF PGRN are associated with worse cognitive performance in all tests investigated even when age, gender *APOE* ε4 status and education (Model 2) are accounted. The associations still remain after adding Aβ$_{1-42}$ and T-tau as added in the model (Model 3) in all tests except in ADNI-Mem, MMSE and CDR-SB.

The analysis was performed excluding CSF PGRN outliers. The same analysis including these outliers yielded similar results (Appendix Table S6). There were two subjects without ADAS-Cog11 score and six subjects without ADAS-Cog13 scores.

*Significant differences.

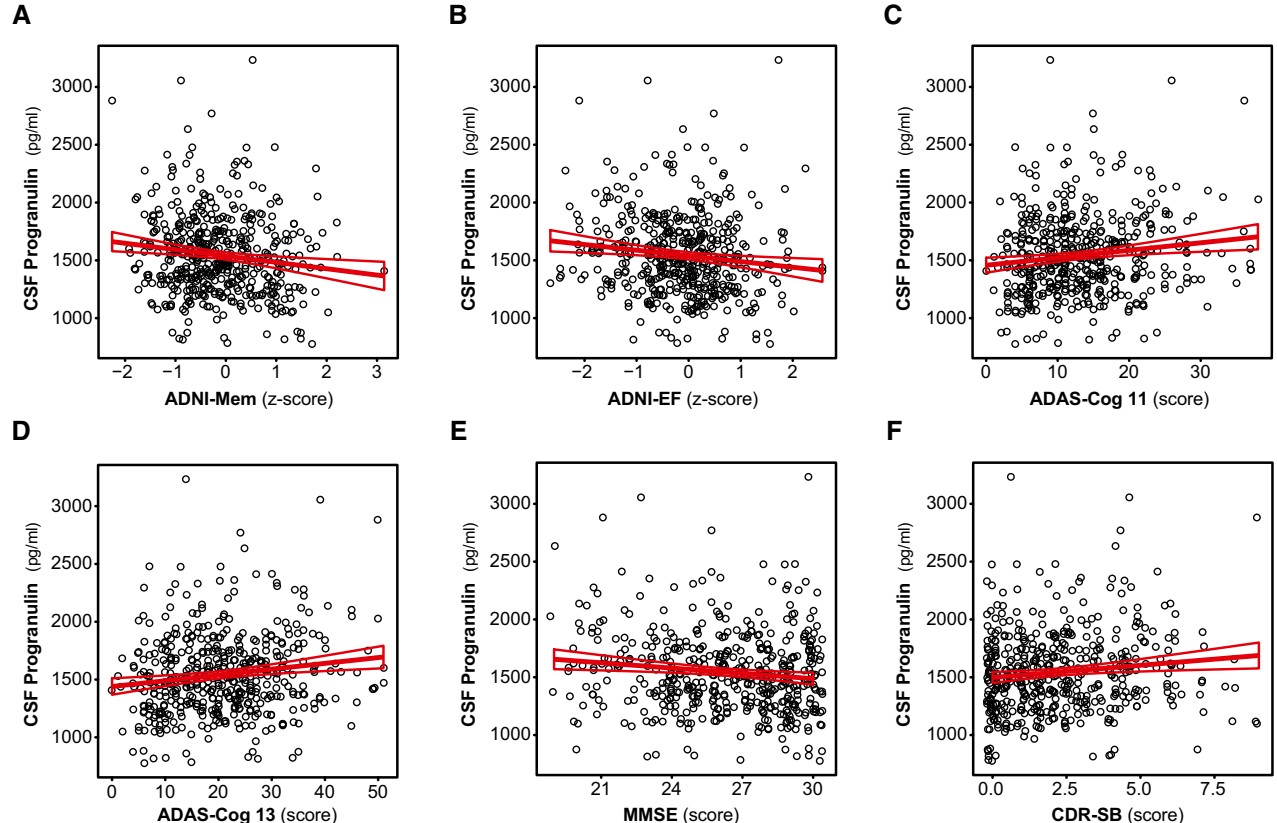

**Figure 4.  CSF PGRN as a function of cognitive function.**

Scatter plots representing the association of CSF PGRN with different cognitive tests. Only the subjects of the Alzheimer's *continuum* group (n = 474) were included. In all tests studied, higher levels of CSF PGRN were associated with worse cognitive performance (namely lower scores in ADNI-Mem, ADNI-EF and MMSE and higher scores in ADAS-Cog11, ADAS-Cog13 and CDR-SB). The analysis and the graphs are excluding PGRN outliers; including them rendered similar results (Appendix Table S6). Each point depicts the value of CSF PGRN and the corresponding cognitive test score of a participant. The solid lines indicate the regression line and the 95% confidence interval (CI) calculated by a linear model (Model 1, unadjusted). Table 5 shows the standardized regression coefficients (β) and the *P*-values calculated by different models. ADAS-Cog, Alzheimer's disease Assessment Scale—cognitive subscale; ADNI-Mem: ADNI memory composite score; ADNI-EF: ADNI executive function composite score; CDR-SB: clinical dementia rating sum of boxes; MMSE, Mini-Mental State Examination.

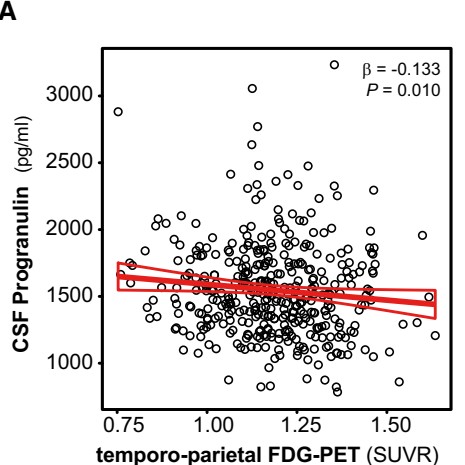

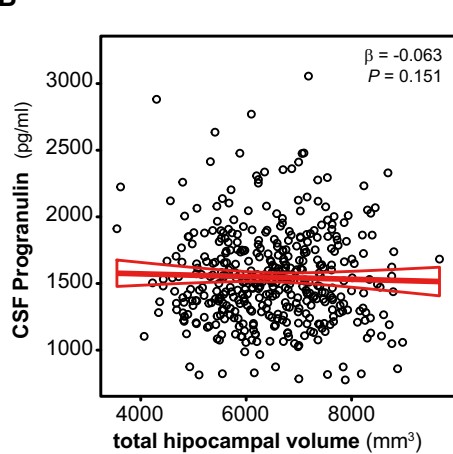

**Figure 5. CSF PGRN as a function of neuroimaging biomarkers.**

A, B    Scatter plots representing the association of CSF PGRN with temporo-parietal FDG-PET uptake (A) and total hippocampal volume (B) within the subjects of the Alzheimer's *continuum* group (n = 474). Each point depicts the value of CSF PGRN and the corresponding neuroimaging biomarker of a participant. The solid lines indicate the regression line and the 95% confidence interval (CI). The regression coefficients (β) and the *P*-values calculated by a linear model adjusted for age, gender, *APOE* ε4 and education. FDG-PET: fludeoxyglucose positron emission tomography; SUVR: standardized uptake value ratio.

protein sTREM2 and with markers of neuronal injury (T-tau) and neurofibrillary tangle degeneration (P-tau$_{181P}$). Given that both PGRN and sTREM2 are predominantly expressed in microglia within the brain, our findings provide further evidence for a crucial role of microglia in modulation of onset and progression of AD. While CSF PGRN cannot serve as a diagnostic marker in AD, PGRN may serve as microglial activity marker that, together with sTREM2, could allow tracking microglial phenotypes not only during the course of the disease but also during therapeutic interventions. Furthermore, since loss of TREM2 and PGRN results in opposite phenotypes (Götzl *et al*, submitted), our findings may also allow to track different activation stages of microglia upon TREM2 or PGRN loss of function. Our study also further reinforces that PGRN is relevant not only for FTD but also for AD.

Although the results in ADAD and late-onset AD consistently show a continuous increase in CSF PGRN while disease progresses, differences between ADAD and late-onset AD were observed. While in ADAD CSF PGRN increased early in the disease (10 years before the expected symptom onset), in late-onset AD, the CSF PGRN levels did not reach a significant increase compared to healthy controls until the mild dementia stage (CDR = 1). Importantly, however, all late-onset AD groups in the Alzheimer's *continuum* had increased CSF PGRN levels compared with the earliest stage of AD, the "Preclinical AD A+TN−" group, which had the lowest CSF PGRN levels. This means that once neurodegeneration and neurofibrillary

tangle degeneration have started (as expressed by the N and T positivity, respectively), CSF PGRN increases. This parallels what occurs in ADAD, where CSF T-tau (and hence, probably neurodegeneration as well) significantly increases 15 years before the symptom onset (Bateman *et al*, 2012; Fagan *et al*, 2014; Suárez-Calvet *et al*, 2016a, b) and this is followed by the later increase in CSF PGRN (EYO = −10, as shown in the present study) and CSF sTREM2 (EYO = −5; Suárez-Calvet *et al*, 2016a,b). Our conclusion is further supported by the fact that CSF PGRN is associated with CSF T-tau, P-tau$_{181P}$, cognitive impairment and temporo-parietal FDG specifically in late-onset AD, but not in SNAP. This may indicate that, although CSF PGRN production may be increased whenever there is neuronal injury, CSF PGRN is specifically coupled to the progression of neurodegeneration in AD. In contrast, CSF PGRN was not associated with hippocampal volume, which may suggest a more complex link between inflammatory-related biomarkers such as CSF PGRN and structural imaging. In fact, CSF sTREM2 was found to be positively associated with grey matter volume in mild cognitive impairment due to AD, despite the fact that CSF sTREM2 increases throughout the early stages of the disease, which was attributed to a possible brain swelling (Gispert *et al*, 2016). CSF PGRN was not either associated with cognitive measurements in ADAD, despite its increase throughout EYO and its clear association with CSF T-tau and P-tau$_{181P}$. This may suggest that in ADAD, CSF PGRN increases early in the disease but remains stable throughout

**Figure 6. Association of CSF PGRN with AD core CSF biomarkers in ADAD (DIAN).**

A–H    Scatter plots representing the associations of CSF PGRN with CSF sTREM2 and each of the AD CSF core biomarkers (T-tau, P-tau$_{181P}$ and Aβ$_{1−42}$) in non-carriers (NC, blue; A, C, E and G) and in mutation carriers (MC, red; B, D, F and H). Each point depicts the value of CSF PGRN and the corresponding biomarker of a subject and the solid lines indicate the regression line and the 95% confidence interval (CI) for each of the groups. The standardized regression coefficients (β) and the *P*-values are shown and were computed using a linear model adjusting for age, gender and *APOE* ε4. The sample contained some outliers (defined as 3 SDs below or above the group mean) of the CSF core markers of AD. The results shown in the figure are excluding these outliers. We also performed the analysis including these outliers which yielded similar results (Appendix Table S8). Aβ$_{1−42}$: amyloid-β 42; T-tau: total tau; P-tau: tau phosphorylated at threonine 181.

▶

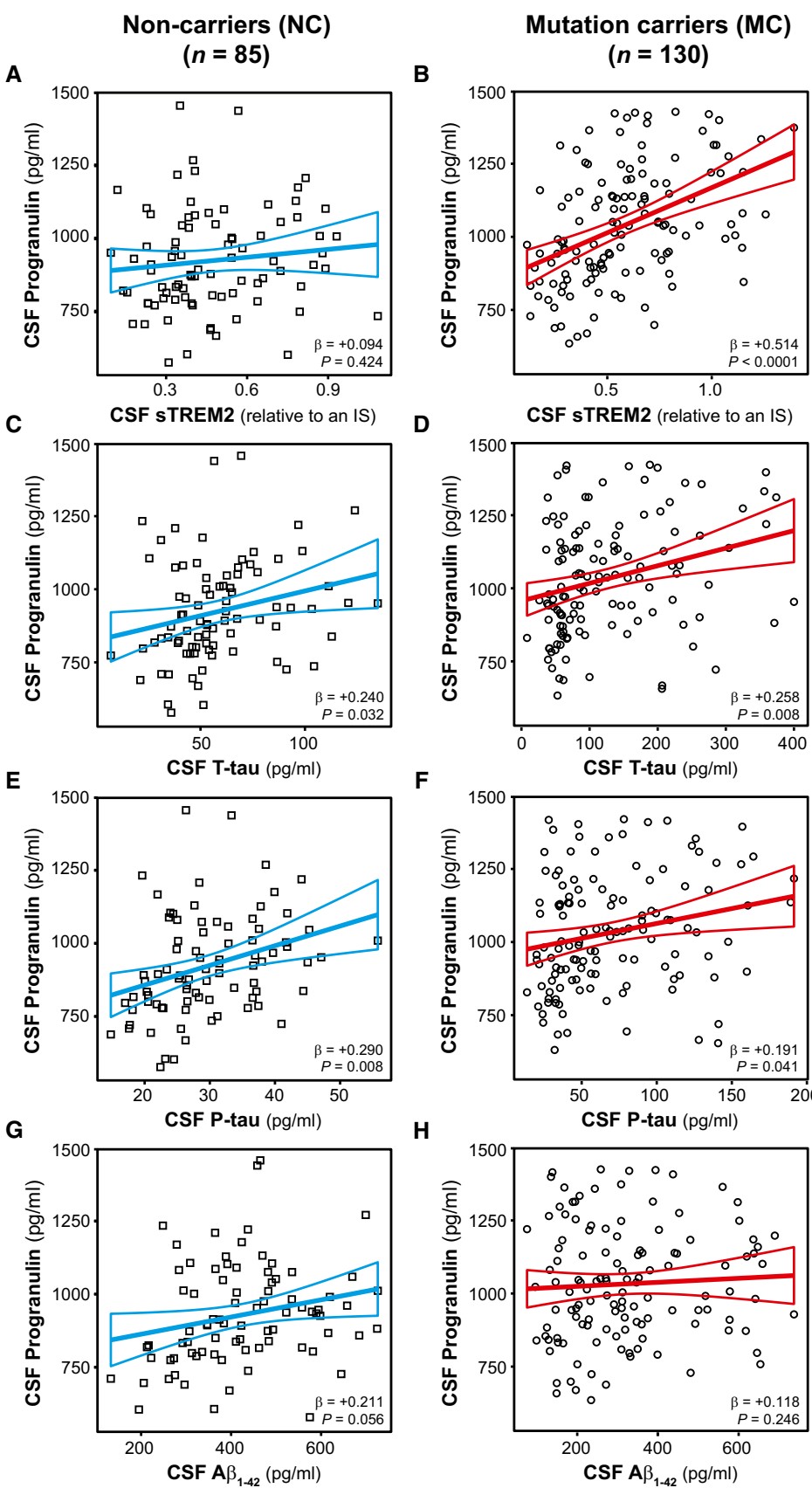

**Figure 6.**

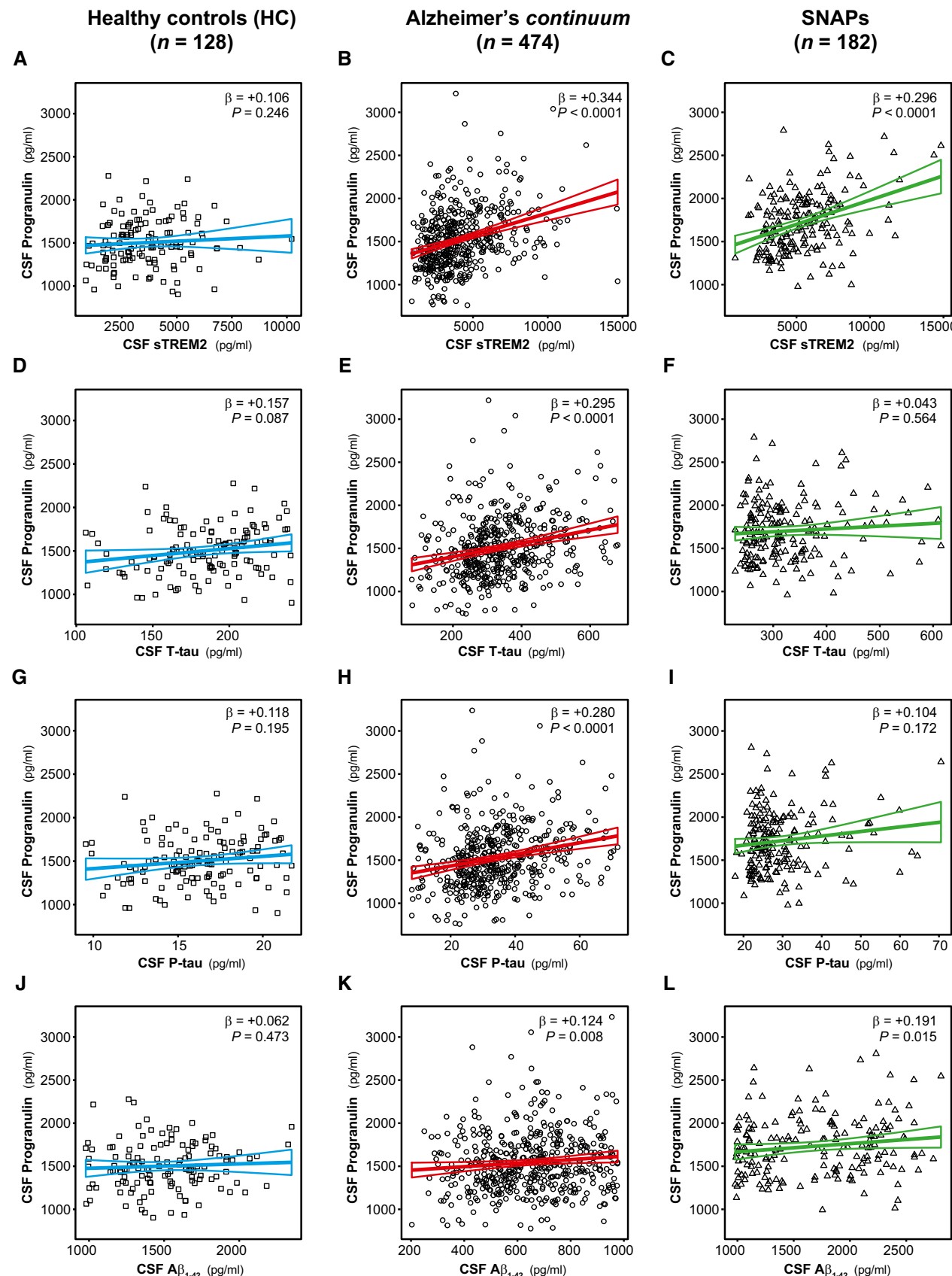

**Figure 7.**

◄

**Figure 7. Association of CSF PGRN with AD core CSF biomarkers in late-onset AD (ADNI).**

A–L   Scatter plots representing the associations of CSF PGRN with CSF sTREM2 and each of the AD CSF core biomarkers (T-tau, P-tau$_{181P}$ and Aβ$_{1-42}$) in healthy controls (blue; A, D, G and J), Alzheimer's *continuum* (red; B, E, H and K) and "suspected non-Alzheimer's pathophysiology (SNAP)" groups (green; C, F, I and L). Each point depicts the value of CSF PGRN and the corresponding biomarker of a subject, and the solid lines indicate the regression line and the 95% confidence interval (CI) for each of the groups. The standardized regression coefficients (β) and the *P*-values are shown and were computed using a linear model adjusting for age, gender and *APOE* ε4. The sample contained some outliers (defined as 3 SDs below or above the group mean) of the CSF core markers of AD, and the analysis including these outliers yielded similar results (Appendix Table S8). The Aβ$_{1-42}$ values used for the association test are those based on an extrapolation curve since the upper technical limit is 1,700 pg/ml. We also tested the associations with Aβ$_{1-42}$ values truncated at the upper technical limit and the result was similar. Aβ$_{1-42}$: amyloid-β 42; T-tau: total tau; P-tau: tau phosphorylated at threonine 181; SNAP: suspected non-Alzheimer's pathophysiology.

the later clinical progression. In ADAD, the causative *PSEN1, PSEN2* and *APP* mutations are such strong driving forces of disease pathology that other factors, such as PGRN, may be less associated with cognition.

In contrast to our findings, previous studies did not detect an increase of CSF PGRN in AD (Nicholson *et al*, 2014; Körtvélyessy *et al*, 2015; Morenas-Rodríguez *et al*, 2015; Wilke *et al*, 2017). This is likely due to the fact that DIAN and ADNI provide a number of samples and an exhaustive clinical data that is unrivalled by any other cohort. In DIAN, we can compare mutation carriers, which are destined to develop AD, with the best controls possible, that is, their non-carriers' siblings. A further advantage of studying ADAD is the use of the concept of estimated years from expected symptom onset (EYO) as a proxy of disease evolution to predict a temporal progression of CSF PGRN, despite the fact that this is a cross-sectional study. This approach has been well validated by studies of the DIAN consortium that have shown that there is a high correlation between the parental age of onset and the mean age at onset of the affected family members (from which EYO is derived) and the actual age of onset (Ryman *et al*, 2014). Finally, in our study we applied an unbiased approach to classify the participants of ADNI, based on biomarker profile (A/T/N) and cognition (C), which made us independent of different diagnostic schemes in different centres (Jack *et al*, 2016a). That allowed us to study those groups that are included in the Alzheimer's *continuum*, from preclinical to mild dementia stages of the disease, and compare them with the evolution of the disease in ADAD as defined by EYO.

The precise mechanism underlying the increase in CSF PGRN during AD has yet to be determined. Since PGRN is highly increased upon activation of microglia (Petkau *et al*, 2010), we speculate that increased CSF PGRN reflects elevated microglial function. It is well established that activation of microglia occurs in AD (Lyman *et al*, 2014; Heneka *et al*, 2015) and, in fact, we and others have previously shown that CSF sTREM2 also increases throughout the evolution of the disease (Heslegrave *et al*, 2016; Piccio *et al*, 2016; Suárez-Calvet *et al*, 2016a,b). A remarkable finding of this study is that higher CSF PGRN levels are associated with higher CSF sTREM2 levels exclusively when disease occurs (either in ADAD MC, late-onset AD or in SNAP) but not in controls. This may indicate that these two proteins are released by microglia adopting a disease-associated signature (Keren-Shaul *et al*, 2017; Krasemann *et al*, 2017). However, it is also possible that these two proteins reflect different populations of microglia co-existing in the disease and may even have opposing effects on disease. While lack of TREM2 locks microglia in a homeostatic state (Mazaheri *et al*, 2017), lack of PGRN locks microglia in a hyperactive state (Götzl *et al*, submitted). Whether these responses are beneficial or detrimental in the disease may depend on the stage of

the disease and needs to be addressed in longitudinal studies. The fact that CSF PGRN is associated with the microglial-derived protein CSF sTREM2 and both are associated with markers of neuronal injury and neurofibrillary tangle degeneration suggests that elevated levels of CSF PGRN and CSF sTREM2 early during the disease reflect a microglial response to neuronal injury. In line with this, recent studies demonstrated that disease-associated *TREM2* variants cause a loss of function suggesting that TREM2 is protective (Kleinberger *et al*, 2014, 2017; Wang *et al*, 2015; Song *et al*, 2018). Similarly, haploinsufficiency of PGRN causes FTD (Baker *et al*, 2006; Cruts *et al*, 2006), and a complete loss of PGRN results in a lysosomal storage disorder with severe neurodegeneration (Smith *et al*, 2012; Götzl *et al*, 2014). Therefore, TREM2 and PGRN may have a protective function at least early during the disease. TREM2 clearly has a cell-autonomous function mediated via DAP12 signalling within microglia (Colonna & Wang, 2016). However, sTREM2, which is released on the plasma membrane, may also have non-cell-autonomous functions. Similarly, PGRN is either targeted directly from the trans-Golgi to lysosomes (where it then may have cell-autonomous functions) or released via the secretory pathway (Götzl *et al*, 2016). In that case PGRN, like sTREM2, may have intrinsic functions within microglia, but both proteins may be able to affect cellular functions in non-microglial cells as well.

Finally, it is important to note that CSF PGRN has no utility as a diagnostic marker in AD since the values overlap considerably between groups. This study has to be interpreted from the perspective of the pathophysiological importance of PGRN in AD, and not as the evaluation of a putative diagnostic biomarker. Taken together, PGRN and sTREM2 may both be increased early in the disease upon an initial response of microglia to first neuronal injury. Their CSF levels may therefore allow conclusions about the functional status of microglia. This is also of specific value for the investigation of novel strategies aiming to modulate microglial activity.

## Materials and Methods

### DIAN participants and study design

The DIAN study is a multicentre study initiated in 2008 that aims to develop a registry of families with known ADAD mutations (namely *PSEN1, PSEN2* or *APP*) and investigate the pathophysiological changes that occur in the different stages of AD (Bateman *et al*, 2012; Morris *et al*, 2012; Suárez-Calvet *et al*, 2016a,b). We measured CSF PGRN from a total of 218 participants, 131 mutation carriers (MC) and 87 non-carriers (NC). All DIAN participants underwent a comprehensive clinical and neuropsychological evaluation (Bateman *et al*, 2012; Storandt *et al*, 2014). CSF collection

follows standard procedures (Fagan et al, 2014; Monserrate et al, 2015), and the measurements of T-tau, P-tau$_{181P}$ and CSF A$\beta_{1-42}$ were performed by the Luminex bead-based multiplexed xMAP technology (INNO-BIA AlzBio3, Innogenetics; Fagan et al, 2014). The estimated years from expected symptom onset (EYO) is calculated as the difference between the participant's age at evaluation and the mean age at onset of all other affected family members (Bateman et al, 2012; Ryman et al, 2014).

### ADNI participants and study design

The ADNI project (http://adni.loni.usc.edu) is a multicentre longitudinal study led by Principal Investigator Michael Weiner and with the main goal to develop and validate biomarkers for subject selection and as surrogate outcome measures in late-onset AD (Weiner et al, 2017). Inclusion and exclusion criteria for the ADNI study were previously described comprehensively (Petersen et al, 2010).

For the present study, we measured CSF PGRN in a cross-sectional sample of the ADNI project consisting of a total of 1,028 individuals. The criteria to select the participants are described in the Appendix Table S9. The CSF PGRN measurements were uploaded to the ADNI database (http://adni.loni.usc.edu) on 16/03/2018, and all the data from ADNI used in this study were downloaded on 21/03/2018.

The CSF core biomarker measurements in ADNI were performed using the electrochemiluminescence immunoassays Elecsys total-tau CSF, phosphor-tau(181P) CSF and β-amyloid (1–42) CSF on a fully automated Elecsys cobas e 601 instrument (Roche) and a single lot of reagents for each of the three measured biomarkers (provided in UPENNBIOMK9.csv file available at http://adni.loni.usc.edu). These immunoassays are for investigational use only. They are currently under development by Roche Diagnostics and not commercially available yet. The Elecsys β-amyloid (1–42) CSF assay measuring range beyond the upper technical limit has not been formally established. Therefore, use of values above the upper technical limit, which are provided based on an extrapolation of the calibration curve, is restricted to exploratory research purposes and is excluded for clinical decision making or for the derivation of medical decision points. The analyte measuring ranges (lower technical limit to upper technical limit) of these assays are the following: 80–1,300 pg/ml for total-tau CSF, 8–120 pg/ml for phosphor-tau(181P) and 200–1,700 pg/ml for Elecsys β-amyloid (1–42) CSF immunoassays. There were 160 values of A$\beta_{1-42}$ that were above the upper technical limit and were truncated to 1,700 pg/ml. There were no samples with T-tau or P-tau$_{181P}$ values above its respective upper technical limit of quantification. For A$\beta_{1-42}$, exploratory measurements are available based on the extrapolation of the calibration curve. The associations of the CSF biomarkers described in the text are using these extrapolated values. The analysis using the A$\beta_{1-42}$ truncated values and those using the extrapolated measurements yielded similar results.

We classified the ADNI participants in a descriptive unbiased approach (Jack et al, 2016a) into two different schemes. First, their biomarker profile, which is defined by the three different pathologic processes that occur in AD and that a biomarker can measure, that is: (i) aggregated Aβ (A, as defined by CSF A$\beta_{1-42}$); (ii) aggregated tau (T, as defined by CSF P-tau$_{181P}$); (iii) neurodegeneration or neuronal injury (N, as defined by CSF T-tau). We binarized each of

the biomarker group into positive (+, abnormal) or negative (−, normal) based on the reported cut-offs for each of the biomarkers (Hansson et al, 2018). Levels below 976.6 pg/ml (A$\beta_{1-42}$) or above 21.8 pg/ml (P-tau$_{181P}$) and 245 pg/ml (T-tau) were categorized as positive (A+, T+, N+, respectively). To reduce the number of groups, we merged the aggregated tau (T) and neurodegeneration (N) groups. If either aggregated tau (T) or neurodegeneration (N) was abnormal (T+ or N+), participants were classified as TN abnormal (TN+). If both aggregated tau (T) and neurodegeneration (N) were normal (T− and N−), participants were classified as TN normal (TN−). Note that in the "healthy control" group, we set the criteria that both aggregated tau (T) and neurodegeneration (N) biomarker profile should be normal, so that we ensure that this group is indeed free of pathology. Importantly, only 54 participants (5.3% of the total sample) displayed discrepancies between the T and N biomarker groups. Second, we classified the participants based on clinical symptoms based on the clinical dementia rating (CDR) global score (Morris, 1993) into cognitively unimpaired (CDR = 0), very mild dementia (CDR = 0.5) and mild dementia (CDR = 1). In the sample studied, there were no participants with moderate or severe dementia (CDR = 2–3). The combination of both the biomarker and the clinical classification rendered 12 different groups that are summarized in Table 3.

Cognition in ADNI participants was assessed by ADNI-Mem, ADNI-EF, ADAS-Cog 11, ADAS-Cog 13, MMSE and CDR-SB. ADNI-Mem and ADNI-EF are both composite scores developed by ADNI. ADNI-Mem is derived from of the following test scores: Rey Auditory Verbal Learning test, Alzheimer's Disease Assessment Schedule-Cognition (ADAS-Cog), MMSE and Logical Memory (Crane et al, 2012). The ADNI-EF is derived from the following tests: Wechsler Adult Intelligence Scale—Revised Digit Symbol Substitution, Digit Span Backwards, Trials A and B, Category Fluency and Clock Drawing (Gibbons et al, 2012).

Hippocampal volumes based on FreeSurfer segmentation and FDG-PET ROI SUVRs derived from meta-analytically regions of AD-associated hypometabolism located within the angular gyrus, posterior cingulate and inferior temporal lobe were downloaded from the ADNI database. The MRI and FDG-PET ROI segmentation has been described in detail previously (Fischl et al, 2002; Landau et al, 2011). Hippocampal volumes were further adjusted for intracranial volume using linear regression, following previous recommendations (Jack et al, 2017).

### Ethical considerations

The study was approved by the institutional review board (IRB) of all participating centres in DIAN and ADNI, as well as our local IRB (LMU). All study participants (or their relatives) provided written informed consent.

### CSF progranulin (PGRN) measurement

CSF PGRN was measured by an ELISA protocol previously established by our group using the MSD Platform (Capell et al, 2011). The ELISA consists of a Streptavidin-coated 96-well plates (MSD Streptavidin Gold Plates, cat. no. L15SA); a biotinylated polyclonal goat anti-human PGRN capture antibody (BAF2420, R&D Systems; 0.2 μg/ml, 25 μl/well); a monoclonal mouse anti-human PGRN

detection antibody (MAB2420, R&D Systems; 0.25 µg/ml, 25 µl/well); and a SULFO-TAG-labelled goat polyclonal anti-mouse IgG secondary antibody (MSD, cat. no. R32AC; 0.5 µg/ml, 25 µl/well). All antibodies were diluted in 0.5% bovine serum albumin (BSA) and 0.05% Tween-20 in PBS buffer (pH = 7.4). Recombinant human PGRN protein (His Tag PGRN—Sino Biological, cat. no. 10826-H08H) was used as a standard (15.6–2,000 pg/ml). In brief, Strepta-vidin-coated 96-well plates were blocked overnight at 4°C in block-ing buffer [0.5% BSA and 0.05% Tween-20 in PBS (pH = 7.4)]. The plates were next incubated with the capture antibody for 1 h at room temperature (RT). They were subsequently washed four times with washing buffer (0.05% Tween-20 in PBS) and incubated for 2 h at RT with the CSF and the internal standard (IS) samples diluted 1:2.5 in assay buffer [0.25% BSA and 0.05% Tween-20 in PBS (pH = 7.4)] or the recombinant human PGRN protein for the standard curve also diluted in assay buffer. CSF samples were randomly distributed across plates and measured in triplicates (DIAN samples) or in duplicates (ADNI samples). The operators were blinded to the clinical information. Plates were again washed four times with washing buffer before incubation for 1 h at RT with the detector antibody. After four additional washing steps, plates were incubated with the secondary antibody for 1 h in the dark. Last, plates were washed four times with washing buffer followed by two washing steps in PBS. The electrochemical signal was devel-oped by adding MSD read buffer T (cat. no. R-92TC) and the light emission measured using the MESO QuickPlex SQ 120. The ELISA showed good accurate results in the spike recovery (98%) experi-ments and minimal measurement variation (CV = 4%) between freeze–thaw cycles (nine cycles).

The measurement of all the DIAN samples was performed in a single day (22/11/2016), and three CSF samples (internal stan-dards, IS) were loaded in all plates. All IS used in this study consisted of pooled CSFs from diagnostic clinical routine leftovers from the Ludwig-Maximilians-Universität München (LMU) Depart-ment of Neurology (Munich, Germany). All patients gave their written consent, and the study was approved by the local IRB. The interplate coefficient of variation (CV) for each of the IS was 3.6, 5.4 and 6.7%. The mean intraplate CV was 2.1%, and all replicate measures had a CV ≤ 15%. The raw values are provided as pg/ml.

The measurements of the ADNI samples were performed on four different days (between the 27/11/2017 and 06/12/2017). Four CSF-IS samples were loaded in all plates. The interplate CV for each of the IS was 4.3, 3.8, 3.4 and 5.4%. The mean intraplate CV was 2.2%, and all replicate measures had a CV ≤ 15%. Given that the ADNI measurements were done in several days, we corrected the raw measurements based on values of the four IS that were loaded on all plates. The concentration of each IS in an individual plate (plate x) was expressed as a percentage of the mean concentration across all plates as follows:

a1 (%) in plate x = [(concentration of IS1 in plate x)/(mean concentration of IS1 in all plates)] × 100
a2 (%) in plate x = [(concentration of IS2 in plate x)/(mean concentration of IS2 in all plates)] × 100
a3 (%) in plate x = [(concentration of IS3 in plate x)/(mean concentration of IS3 in all plates)] × 100
a4 (%) in plate x = [(concentration of IS4 in plate x)/(mean concentration of IS4 in all plates)] × 100

The mean of the percentages (Ax) for all the IS (a1, a2, a3, a4) in plate x was calculated, and the following correction factor was computed for each individual plate:

Correction factor for plate x = 100/Ax

The raw values were multiplied by the correction factor of the corresponding plate; the corrected values are provided as the vari-able "MSD_PGRNCORRECTED" in the ADNI database. Due to the low interplate CV, the corrected and raw PGRN values in the ADNI sample are highly correlated (Spearman ρ = 0.985; *P* < 0.0001).

The mean CSF PGRN level of our MSD-based assay (1,007 pg/ml) was similar to that published with other assays (Huchtemann *et al*, 2015; Zhou *et al*, 2015a; Berghoff *et al*, 2016; Meeter *et al*, 2016; Kimura *et al*, 2017; Schreiber *et al*, 2018), albeit lower than those measured by the more widely used commercial assay from Adipogen (Ghidoni *et al*, 2008; De Riz *et al*, 2010; Vercellino *et al*, 2011; Nicholson *et al*, 2014; Morenas-Rodríguez *et al*, 2015; Feneberg *et al*, 2016; Molgaard *et al*, 2016; Willemse *et al*, 2016; Wilke *et al*, 2017). We therefore compared our MSD-based ELISA with that from Adipogen (cat. no. AG-45A-0018YEK-KI01, Seoul, Korea). We prepared 39 different CSF pool samples from leftovers of the LMU Department of Neurology, as described above. We measured these CSF samples simultaneously in both the MSD-based and the Adipogen assays, following the manufacturer's instructors, and each sample in duplicate. The standard curves for each of the assays are shown in Appendix Fig S1A and B. As expected, the mean levels of CSF PGRN were lower in the MSD-based assay (mean = 1430 pg/ml, SD = 220) compared to the Adipogen assay (mean = 4,400 pg/ml, SD = 900), but they were highly correlated between them (Spearman ρ = 0.74; *P* < 0.0001; Appendix Fig S1C).

**CSF sTREM2 measurement**

CSF sTREM2 measurements from the DIAN study were previously reported (Suárez-Calvet *et al*, 2016a,b) and are expressed relative to an IS. Measurements of the ADNI samples were done with the same ELISA protocol with minor changes. Briefly, the assay is based on the MSD platform and it is comprehensively described in previous publications (Kleinberger *et al*, 2014; Suárez-Calvet *et al*, 2016a,b). The assay consists of a Streptavidin-coated 96-well plates (MSD Streptavidin Gold Plates, cat. no. L15SA); a biotinylated polyclonal goat IgG anti-human TREM2 antibody (R&D Systems, cat. no. BAF1828; 0.25 µg/ml, 25 µl/well) as capture antibody, which is raised against amino acids 19-174 of human TREM2; a monoclonal mouse IgG anti-human TREM2 antibody (Santa Cruz Biotechnology, B-3, cat. no. sc373828; 1 µg/ml, 50 µl/well) as a detection antibody, which is raised against amino acids 1–160 of human TREM2; and a SULFO-TAG-labelled goat polyclonal anti-mouse IgG secondary anti-body (MSD, cat. no. R32AC; 0.5 µg/ml, 25 µl/well). All antibodies were diluted in 1% BSA and 0.05% Tween-20 in PBS buffer (pH = 7.4). Recombinant human TREM2 protein (Hölzel Diagnos-tika, cat. no. 11084-H08H), corresponding to the extracellular domain of human TREM2 (amino acids 19–174), was used as a stan-dard (62.5–8,000 pg/ml). In brief, Streptavidin-coated 96-well plates were blocked overnight at 4°C in blocking buffer [3% bovine serum albumin (BSA) and 0.05% Tween-20 in PBS (pH = 7.4); 300 µl/well]. The plates were next incubated with the capture antibody for

1 h at RT. They were subsequently washed four times with washing buffer (200 μl/well; 0.05% Tween-20 in PBS). Thereafter, the recombinant human TREM2 protein (standard curve), the blanks, and the CSF and the internal standard (IS) samples (duplicates; dilution factor: 4) were diluted in assay buffer [0.25% BSA and 0.05% Tween-20 in PBS (pH = 7.4)] supplemented with protease inhibitors (Sigma; Cat. # P8340) and incubated (50 μl/well) for 2 h at RT. This dilution was previously selected because it showed the best recovery and linearity performance (Kleinberger *et al*, 2014). Plates were again washed four times with washing buffer before incubation for 1 h at RT with detection antibody. After four additional washing steps, plates were incubated with SULFO-tag conjugated secondary antibody for 1 h in the dark at RT. Last, plates were washed four times with washing buffer followed by two washing steps in PBS. The electrochemical signal was developed by adding 150 μl/well MSD read buffer T (cat. no. R-92TC) and the light emission measured using the MESO QuickPlex SQ 120. Raw values are provided as pg/ml.

All CSF samples were distributed randomly across plates, measured in duplicate and simultaneously to CSF PGRN (i.e. between 27/11/2017 and 06/12/2017). The mean intraplate CV was 3.1%, and all duplicate measures had a CV < 15%. Alike the CSF PGRN ELISA, four CSF IS samples were loaded in all plates. The interplate CV for each of the IS was 11.4, 12.2, 10.5 and 7.1%. We corrected the raw measurements based on values of the four IS that were loaded on all plates in a similar manner as in CSF PGRN measurements. Consequently, the corrected values were used and are available in the ADNI database as variables "MSD_sTREM2CORRECTED".

## Statistical analysis

In both DIAN and ADNI samples, we only included in our study participants that had the following data available: age, gender and the three AD CSF core biomarkers ($A\beta_{1-42}$, T-tau, P-tau$_{181P}$). Within each of the samples, we determined the CSF PGRN outliers, as defined as values differing 3 standard deviations from the mean (3 outliers in the DIAN sample and 11 in the ADNI sample). In order to rule out that our results are not driven by extreme values, all the analysis described in the main text are performed without these outliers. Nevertheless, including them did not affect the main results (as shown in the main text and in Appendix Tables S1, S3, S5, S6 and S8). In the DIAN sample, CSF PGRN followed a normal distribution as assessed by visual inspection of histogram and after testing by Kolmogorov–Smirnov test ($P = 0.062$). In contrast, CSF PGRN in ADNI did not follow a normal distribution (Kolmogorov–Smirnov test, $P = 0.0001$). After a $\log_{10}$ transformation, it followed a normal distribution (Kolmogorov–Smirnov test, $P = 0.200$). All the statistical analyses in the ADNI sample were hence performed with the $\log_{10}$-transformed values.

The data from the DIAN sample (Data Freeze 9) were studied in a similar approach to that we previously applied (Suárez-Calvet *et al*, 2016a,b). In brief, comparisons of demographic, clinical and biochemical data between NC and MC were performed by Pearson's chi-square tests or *t*-tests, as appropriate. A linear regression analysis was used to test the effect of the mutation status, age, gender and *APOE* ε4 status in the levels of CSF PGRN (Appendix Table S1). In additional linear effects models, we compared the levels of CSF

PGRN between carriers of the three mutated genes (i.e. *PSEN1, PSEN2* and *APP*).

In order to test how CSF PGRN levels change as a function of EYO (as shown in Table 2 and Fig 2), we used a similar approach to that previously published (Bateman *et al*, 2012; Fagan *et al*, 2014; Suárez-Calvet *et al*, 2016a,b). We constructed a linear mixed model with mutation status, EYO (and its interaction with mutation status) and gender as fixed effects and family affiliation as random effect. Next, we performed a polynomial regression analysis including EYO quadratic (EYO$^2$) and cubic (EYO$^3$) terms and their interactions with mutation status. We determined the model that best fitted the data by forward selection of the predictors, and the final model was chosen based on the Akaike information criterion (AIC; a lower AIC indicating a better fit; see Appendix Table S2). For CSF PGRN, the linear model (first-order EYO) showed the best fit. In previous studies, we used linear mixed-effects models with family membership as a random effect to adjust for differences in biomarker levels between families. Here, we found no significant differences in CSF PGRN levels between families ($F_{93,114} = 1.04$, $P = 0.426$) and we therefore used simple linear regression in all the following analysis.

We computed the estimated levels of CSF PGRN at each 5-year interval of EYO based on the established regression models, and we determined the group differences between MC and NC for each 5-year EYO interval by *t*-tests, as done in previous DIAN studies (Bateman *et al*, 2012; Fagan *et al*, 2014; Suárez-Calvet *et al*, 2016a,b). Comparisons between MC and NC were restricted at EYO ranging from −25 to +10, due to the low number of subjects at more extreme values of EYO.

In order to represent the progression of CSF PGRN throughout the evolution of the disease and compare it with other biomarkers and cognitive measures, we followed the approach of previous DIAN studies (Bateman *et al*, 2012; Suárez-Calvet *et al*, 2016a,b); that is, the predicted difference between MC and NC at each EYO generated by the same final linear mixed-effects model described above was divided by the standard deviation of clinical, cognitive, imaging and biochemical measures of the pooled sample, so that all variables were in a standardized and comparable scale (Fig 2B, Appendix Table S2). These figures were built with SAS software (SAS Institute).

We compared CSF PGRN levels between NC and MC in different clinical stages (determined by global CDR) in an analysis of covariance (ANCOVA) controlling for age, gender, *APOE* ε4 and education, followed by *post hoc* least significant difference (LSD) for pairwise comparisons. Individuals falling into CDR = 1 to CDR = 3 were grouped together due to the low number of subjects in these stages.

In ADNI, demographic, clinical and biochemical data group comparisons were performed by one-way ANOVA or Pearson's chi-square tests, as appropriate. To test whether CSF PGRN changed across the Alzheimer's *continuum*, we applied an ANCOVA including the biomarker-defined stages of AD, gender and *APOE* ε4 status as fixed effects and age as covariate, followed by Bonferroni corrected pair-wise *post hoc* comparisons. An additional analysis was performed adding the *GRN rs5848* genotype as a covariate. A similar approach was used for comparisons between biomarker categories shown in Fig EV1A.

To study the associations between CSF PGRN and cognitive and functional scores, we applied three linear regression models. Model

### The paper explained

#### Problem

Alzheimer's disease (AD) is the most frequent neurodegenerative disorder. Besides β-amyloid and tau deposits, all AD brains invariably show neuroinflammatory symptoms associated with microgliosis. Progranulin (PGRN), a secreted protein whose loss of function is associated with frontotemporal lobar degeneration and neuronal ceroid lipofuscinosis, is also genetically involved in AD. In the brain, PGRN is predominantly expressed in microglia. In this study, we investigated PGRN concentration in the cerebrospinal fluid (CSF) of two of the largest and best characterized AD patient cohorts, namely the Dominant Inherited Alzheimer's Disease Network (DIAN), which includes families with autosomal dominant AD (ADAD), and the Alzheimer's Disease Neuroimaging Initiative (ADNI), which studies the evolution of late-onset AD.

#### Results

In carriers of ADAD causing dominant mutations (DIAN), we found that CSF PGRN increased 10 years before the expected symptom onset. In late-onset AD (ADNI), higher CSF PGRN was associated with more advanced disease stages as well as cognitive impairment. CSF PGRN was associated with levels of proteolytically generated soluble TREM2 (sTREM2), a protein described to be a central regulator of microglial function and known to increase 5 years before the expected symptom onset. Importantly, the association between CSF sTREM2 and CSF PGRN was only observed when there was underlying pathology, but not in controls.

#### Impact

CSF PGRN together with CSF sTREM2 may serve as a microglia activity marker in AD and could be used to prove target engagement in clinical trials aiming to modulate microglial activity.

1 was unadjusted, Model 2 included age, gender, *APOE* ε4 and education as covariates, and Model 3 included the former covariates and also CSF $A\beta_{1-42}$ and CSF T-tau.

The association between CSF PGRN and temporo-parietal FDG-PET and total hippocampal volume was tested with a linear regression adjusted for age, gender, *APOE* ε4 and education. Analyses including hippocampal volume were adjusted for intracranial volume.

The diagnostic value of CSF PGRN to discriminate AD (ADAD or late-onset AD) from controls was tested with a receiver operating characteristic (ROC) analysis. We computed areas under the curve (AUC), and we tested whether they were significantly different from the null hypothesis that the AUC equals 0.50, which corresponds to a random test.

Finally, the association between CSF PGRN and CSF sTREM2 and the CSF core biomarkers for AD (T-tau, P-tau$_{181P}$, $A\beta_{1-42}$) was studied with a linear model adjusted for age, gender and *APOE* ε4 status. The analysis was performed stratifying for the mutation status (DIAN) or classifying the subjects in the healthy controls, Alzheimer's *continuum* or SNAP categories (ADNI). The standardized regression coefficients (β) are reported. In order to rule out that the associations were driven by extreme values, we performed the analysis both including and excluding biomarker outliers (defined as AD CSF core biomarkers 3 standard deviations below or above the group mean) and the analysis yielded similar results (Appendix Table S8).

Statistical analysis was performed in SPSS IBM, version 20.0, statistical software and the free statistical software R (http://www.r-project.org/). Figures were built using GraphPad Prism or R. All tests were two-tailed, with a significance level of α = 0.05.

**Expanded View** for this article is available online.

## Acknowledgements

We would like to thank Tammie Benzinger, Krista Moulder, Peter Wang, Chengjie Xiong, Michael Donohue, Thomas Montine, Jihn K. Hsiao, Jacob Alexander and all the researchers in the DIAN and in the ADNI initiatives. We also thank Brigitte Nuscher and Nicole Exner for technical assistance and José Luís Molinuevo and Nicholas Ashton for critically reading the manuscript and helpful discussion. This work was supported by the Deutsche Forschungsgemeinschaft (DFG) within the framework of the Munich Cluster for Systems Neurology (EXC 1010 SyNergy), a DFG funded Koselleck Project (HA1737/16-1 to C.H.) and the FTD Biomarker Award. Data collection and sharing for this project were supported by The Dominantly Inherited Alzheimer's Network (DIAN, UF1AG032438) funded by the National Institute on Aging (NIA), the German Center for Neurodegenerative Diseases (DZNE), Raul Carrea Institute for Neurological Research (FLENI), Partial support by the Research and Development Grants for Dementia from Japan Agency for Medical Research and Development, AMED, and the Korea Health Technology R&D Project through the Korea Health Industry Development Institute (KHIDI). This manuscript has been reviewed by DIAN study investigators for scientific content and consistency of data interpretation with previous DIAN study publications. We acknowledge the altruism of the participants and their families and contributions of the DIAN research and support staff at each of the participating sites for their contributions to this study. This work was also supported by grants from the HHS | NIH | National Institute on Aging (NIA) (R01AG044546, RF1AG053303, R01AG058501 and U01AG058922). YD is supported by an HHS | NIH | National Institute of Mental Health (NIMH) institutional training grant (T32MH014877). LP was supported by a grant from the Fondazione Italiana Sclerosi Multipla (FISM) (FISM 2017/R/20). EM was supported by a grant from the Ad-Hoc Committee for Young Neurologist (Spanish Society of Neurology) and Health Institute Carlos III (funding programme for the mobility of the researchers).

## Author contributions

MS-C, AC, EM-R, KF, GK, EE, YD and LP performed the experiments. MS-C, AC, MAAC, EM-R, NF, YD, LP, CC, CMK, ME and CH analysed and interpreted the data. YD, LP, CMK and CC involved in extracting the genetic data. KP, RJB, AMF, JCM, JL, AD, MJ, CLM, MNR, JMR, LMS, JQT and MW contributed with patient samples and/or data. MS-C, ME and CH designed the study and wrote the manuscript. All authors critically reviewed and approved the final manuscript.

## Conflict of interest

C.H. collaborates with DENALI Therapeutics. The remaining authors declare that they have no conflict of interest.

## For more information

All data of the ADNI study (including the CSF PGRN and sTREM2 reported in this study) are publicly available in http://adni.loni.usc.edu/.

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
