## [Review Process File · EMBO Molecular Medicine]

CSF progranulin increases in the course of Alzheimer's disease and is associated with sTREM2, neurodegeneration and cognitive decline

Marc Suárez-Calvet, Anja Capell, Miguel Ángel Araque Caballero, Estrella Morenas-Rodríguez, Katrin Fellerer, Nicolai Franzmeier, Gernot Kleinberger, Erden Eren, Yuetiva Deming, Laura Piccio, Celeste M. Karch, Carlos Cruchaga, Katrina Paumier, Randall J. Bateman, Anne M. Fagan, John C. Morris, Johannes Levin, Adrian Danek, Mathias Jucker, Colin L. Masters, Martin N. Rossor, John M. Ringman, Leslie M. Shaw, John Q. Trojanowski, Michael Weiner, Michael Ewers, Christian Haass for the Dominantly Inherited Alzheimer Network and for the Alzheimer's Disease Neuroimaging Initiative

Review timeline:

Submission date:	11 October 2018
Editorial Decision:	15 October 2018
Revision received:	19 October 2018
Accepted:	22 October 2018

Editor: Céline Carret

Transaction Report:

Please note that the manuscript was previously reviewed at another journal and the reports were taken into account in the decision making process at EMBO Molecular Medicine. Since the original reviews are not subject to EMBO's transparent review process policy, the reports and author response cannot be published.

Corresponding Author Name: MARC SUÁREZ-CALVET and CHRISTIAN HAASS

Journal Submitted to: EMBO MOLECULAR MEDICINE

Manuscript Number: